# Activity-dependent modulation of synapse-regulating genes in astrocytes

**Isabella Farhy-Tselnicker[1]\*[†], Matthew M Boisvert[1][‡], Hanqing Liu[2,3], Cari Dowling[1], Galina A Erikson[4], Elena Blanco-Suarez[1][§], Chen Farhy[5], Maxim N Shokhirev[4], Joseph R Ecker[2,6], Nicola J Allen[1]\***

[1]Molecular Neurobiology Laboratory, The Salk Institute for Biological Studies, La Jolla, United States; [2]Genomic Analysis Laboratory, The Salk Institute for Biological Studies, La Jolla, United States; [3]Division of Biological Sciences, University of California San Diego, La Jolla, United States; [4]Razavi Newman Integrative Genomics and Bioinformatics Core, The Salk Institute for Biological Studies, La Jolla, United States; [5]Sanford Burnham Prebys Medical Discovery Institute, La Jolla, United States; [6]Howard Hughes Medical Institute, The Salk Institute for Biological Studies, La Jolla, United States

**\*For correspondence:**
ifarhy@bio.tamu.edu (IF-T);
nallen@salk.edu (NJA)

**Present address:** [†]Department of Biology, Texas A&M University, College Station, United States; [‡]Jungers Center for Neuroscience Research, Department of Neurology, Oregon Health and Science University, Portland, United States; [§]Department of Neuroscience, Thomas Jefferson University Hospital for Neuroscience, Philadelphia, United States

**Competing interest:** The authors declare that no competing interests exist.

**Abstract** Astrocytes regulate the formation and function of neuronal synapses via multiple signals; however, what controls regional and temporal expression of these signals during development is unknown. We determined the expression profile of astrocyte synapse-regulating genes in the developing mouse visual cortex, identifying astrocyte signals that show differential temporal and layer-enriched expression. These patterns are not intrinsic to astrocytes, but regulated by visually evoked neuronal activity, as they are absent in mice lacking glutamate release from thalamocortical terminals. Consequently, synapses remain immature. Expression of synapse-regulating genes and synaptic development is also altered when astrocyte signaling is blunted by diminishing calcium release from astrocyte stores. Single-nucleus RNA sequencing identified groups of astrocytic genes regulated by neuronal and astrocyte activity, and a cassette of genes that show layer-specific enrichment. Thus, the development of cortical circuits requires coordinated signaling between astrocytes and neurons, highlighting astrocytes as a target to manipulate in neurodevelopmental disorders.

## Introduction

Synapses are points of contact where electrochemical signals are transferred between neurons in a given circuit (*Petzoldt and Sigrist, 2014*; *Südhof, 2018*). Synapse development occurs in several molecularly and functionally defined stages, which include initiation, maturation, and pruning steps prior to stabilization and establishment of mature circuits. In the rodent cortex, this occurs over the period of the first postnatal month, initiating at around postnatal day (P) 7, peaking at P14, and stabilizing towards P28 (*Blue and Parnavelas, 1983a*; *Blue and Parnavelas, 1983b*; *Farhy-Tselnicker and Allen, 2018*; *Li et al., 2010*). Synaptic deficits, for example, caused by mutations in synapse-related genes, are associated with developmental disorders such as autism spectrum disorder and epilepsy (*Lepeta et al., 2016*). Therefore, understanding how synaptic development is regulated will provide important insights into how circuits form and function in health and are altered in disease.

The majority of synapses in the mammalian cortex are contacted by astrocytes, a type of glia and key regulators of circuit development and function (*Allen, 2013*; *Batool et al., 2019*; *Bernardinelli et al., 2014a*; *Genoud et al., 2006*). Astrocytes express many genes encoding proteins that regulate distinct stages of synapse formation and maturation (*Baldwin and Eroglu, 2017*). For example, thrombospondin family members induce formation of structurally normal but functionally silent

synapses (*Christopherson et al., 2005*; *Eroglu et al., 2009*). Glypicans induce the formation of active synapses by recruiting GLUA1 to the postsynaptic side (*Allen et al., 2012*; *Farhy-Tselnicker et al., 2017*) and chordin-like 1 induces synapse maturation by recruiting GLUA2 to the postsynaptic side (*Blanco-Suarez et al., 2018*). These and other signals have been identified using in vitro cell culture approaches and analyzed individually across ages and brain regions in vivo. Yet, a systematic analysis of their expression patterns in a complete circuit in vivo, which would reveal distinct regulatory roles for specific synaptic connections, has not been attempted.

The mouse visual circuit is a well-characterized and powerful model to investigate the role of astrocytes in regulating different stages of synaptogenesis. Visual information perceived by the retina is relayed to the visual cortex (VC) via the neurons of the thalamic lateral geniculate nucleus (LGN). Before eye opening (from birth to ~P12 in mice), spontaneous retinal activity evokes correlated cortical responses (*Gribizis et al., 2019*; *Hanganu et al., 2006*) that are important for the correct establishment of thalamocortical synapses (*Cang et al., 2005*). Eye opening marks a step towards synapse maturation in the VC, with the appearance of visually evoked neuronal responses across the retinal-LGN-VC circuit (*Espinosa and Stryker, 2012*; *Hooks and Chen, 2006*; *Hooks and Chen, 2020*). Perturbing this process by methods of visual deprivation, such as dark rearing, delays both synaptic (*Albanese et al., 1983*; *Desai et al., 2002*; *Freire, 1978*; *Funahashi et al., 2013*; *Hsu et al., 2018*; *Ishikawa et al., 2014*; *Ko et al., 2014*; *Majdan and Shatz, 2006*; *Tropea et al., 2006*) and astrocyte maturation (*Müller, 1990*; *Stogsdill et al., 2017*).

The VC's glutamatergic neurons are arranged in spatially defined layers, with distinct transcriptomic, functional, and connectivity profiles (*Bannister, 2005*; *Douglas and Martin, 2004*). For example, neurons in layers 1 and 4 receive input from the LGN, while layer 5 neurons, the main output to subcortical regions, send their dendrites to layer 1, where they receive input from both local and subcortical projections. Recent work has shown that cortical astrocytes are also spatially arranged in diverse populations (*Batiuk et al., 2020*; *Bayraktar et al., 2020*; *John Lin et al., 2017*; *Lanjakornsiripan et al., 2018*), in line with evidence from other brain regions showing astrocyte heterogeneity (*Chaboub and Deneen, 2012*; *Chai et al., 2017*; *Khakh and Deneen, 2019*; *Oberheim et al., 2012*; *Rusnakova et al., 2013*; *Schitine et al., 2015*). However, whether this astrocyte diversity is reflected in their regulation of synapses across the distinct layers of the VC is unknown. To understand how the diverse astrocytic signals act together to regulate formation of a complete circuit, it is important to determine both temporal and spatial expression patterns of astrocytic synapse-regulating genes in vivo as well as identify the mechanisms that control their level of expression and the subsequent effect on synapse formation.

Here, we use RNA sequencing and in situ hybridization to obtain the developmental transcriptome of astrocytes in the mouse VC in vivo. We find that astrocyte synapse-regulating genes display differential temporal and spatial expression patterns, which correspond to stages of synapse initiation and maturation. Furthermore, we find that developmental regulation of these genes, namely glypican 4 and chordin-like 1, depends on thalamocortical neuronal activity, with additional regulation by astrocyte IP3R2-dependent $Ca^{2+}$ activity. Manipulating either neuronal or astrocytic activity leads to shifts in synaptic development and maturation. Finally, single-nucleus RNA sequencing analysis reveals diverse populations of astrocytes in the developing VC and identifies novel groups of genes that are regulated by neuronal and astrocyte activity. These findings demonstrate how astrocyte expression of synapse-regulating genes is controlled during development, and how synapse maturation is dependent on neuron-astrocyte communication. These data further provide an important resource for future studies of astrocyte development and astrocyte regulation of synapse formation.

## Results

To address key questions regarding astrocyte regulation of synapse development, this study has three major goals. First, to characterize the spatio-temporal profile of astrocyte development during distinct stages of cortical synaptogenesis at the transcriptomic level. Second, to investigate the mechanisms that influence spatio-temporal expression levels of select astrocytic synapse-regulating genes, focusing on neuronal and astrocyte activity. Third, to determine the global dependence of the astrocyte transcriptome on neuronal and astrocyte activity using an unbiased RNA sequencing approach. All analyses were performed in the mouse VC to enable comparison of these metrics within a defined circuit, providing a blueprint for future analysis of the functional roles of astrocytes in synaptic development.

## Developmental profiling of the astrocyte transcriptome in the postnatal VC

Astrocytes appear in the cortex at birth, and migrate, proliferate, and mature throughout the first postnatal month (*Ge et al., 2012*), coincidently with the stages of synapse development. To determine the transcriptomic profile of astrocytes at these stages (P7, P14, P28, and adult, P120), we used the astrocyte Ribotag mouse model to isolate mRNA for bulk RNA sequencing (B6N.129-Rpl22$^{tm1.1Psam}$/J crossed to B6.Cg-Tg(Gfap-cre)73.12Mvs/J – labeled Astrocyte-RiboTag) (*Figure 1*, *Figure 1—figure supplement 1*, *Figure 1—source data 1*; *Boisvert et al., 2018*; *Chai et al., 2017*). In this mouse line, cre recombinase drives the expression of an HA-tagged ribosomal subunit, enabling purification of cell-type-specific ribosomes and associated mRNA for analysis by RNA sequencing (*Figure 1A and B*). We found a significant enrichment in astrocytic genes over other cell types in the mRNA isolated by HA immunopurification (IP; labeled Astro) compared to total VC mRNA (input; IN; *Figure 1—figure supplement 1A*). Furthermore, immunostaining analysis of VC sections at P28 shows a high overlap between the HA ribosome tag and the astrocyte marker S100β, but not with other cell-type markers (*Figure 1—figure supplement 1B-D*), with more than 95% of S100β-positive cells also positive for the HA tag, suggesting high astrocyte coverage of our mRNA isolation, consistent with our previous analysis in the adult (*Boisvert et al., 2018*). The complete RNA sequencing dataset is available in a searchable format online (http://igc1.salk.edu:3838/astrocyte_transcriptome/).

To assess if there are broad changes in the transcriptomic profiles of astrocytes across development, we performed principal component analysis (PCA; *Figure 1C*). This showed that P7 and P14 astrocytes form distinct clusters, while P28 and P120 astrocytes cluster together. To investigate this further, we analyzed the number of differentially expressed genes (DEGs; fragments per kilobase of exon per million mapped fragments [FPKM] > 1; false discovery rate [FDR] < 0.05) between each age group. DEGs are classified into total genes (all changes detected), genes that are expressed by astrocytes (IP/input > 0.75), and genes that are enriched in astrocytes (IP/input > 3; *Figure 1D*, for definitions, see also *Boisvert et al., 2018*). The largest number of DEGs is between P7 and P28 (~6000 total genes), and smallest numbers between P14 and P28 (~1000 total genes), and P28 and P120 (~2000 total genes). Analysis of astrocyte-enriched genes (IP/input > 3) showed that ~60% of all astrocyte-enriched genes are significantly changed from P7 to P14, while only ~20% are changing between P28 and P120 (*Figure 1E*). This shows that most changes in astrocyte gene expression are occurring between the first and second postnatal weeks, a time of transition from synapse formation to synapse maturation, and in the VC, from spontaneous to visually evoked neuronal activity.

To determine the different astrocyte functions at each age, we focused on the astrocyte-enriched gene lists (~1000 total genes FPKM >1, astro IP/input > 3). To identify common genes expressed at all ages, as well as astrocyte-enriched genes that are most highly expressed at each age, we performed Gene Ontology (GO) analysis, focusing on Biological Process (BP) terms (*Figure 1F–H*, *Figure 1—figure supplement 1F and G*). This showed that genes common to all ages are enriched in GO terms related to cholesterol processing and serine synthesis, confirming the previously established important role of astrocytes in regulating brain cholesterol (*Orth and Bellosta, 2012*; *Figure 1G*). Analysis of GO terms unique to each age showed that at P7 astrocyte genes are enriched in GO terms related to cortical development, while at P14 astrocyte genes are enriched in GO terms related to Wnt and BMP signaling pathways. Conversely, adult astrocytes (P120) are enriched in terms related to regulation of extracellular matrix assembly and contact inhibition (*Figure 1H*). In all we found both the number of genes and the GO terms common to all ages constitute about 50% of all terms identified for each age, while genes and GO terms unique to each age consist less than 10% of all terms (*Figure 1—figure supplement 1F and G*). These results demonstrate that astrocytes perform their core functions across all developmental stages similarly, with several age-specific functions that turn on and off depending on the developmental stage and the environment in the VC.

An important analysis to be performed with this dataset is determining the temporal expression changes of known astrocyte genes, for example, to identify potential astrocyte markers that are either global or age specific. We found that *Apoe* and *Cst3* are the most highly expressed genes in astrocytes at all ages (peak FPKM ~10,000; *Figure 1F*, *Figure 1—source data 1*), while amongst highly expressed astrocyte genes (FPKM >100) *Lars2* is the most astrocyte-enriched gene at all ages (astro IP/ input ~40; *Figure 1—figure supplement 1E*). These genes are highly expressed in adult astrocytes across multiple brain regions (*Morel et al., 2017*), thus can be potentially utilized to mark or target

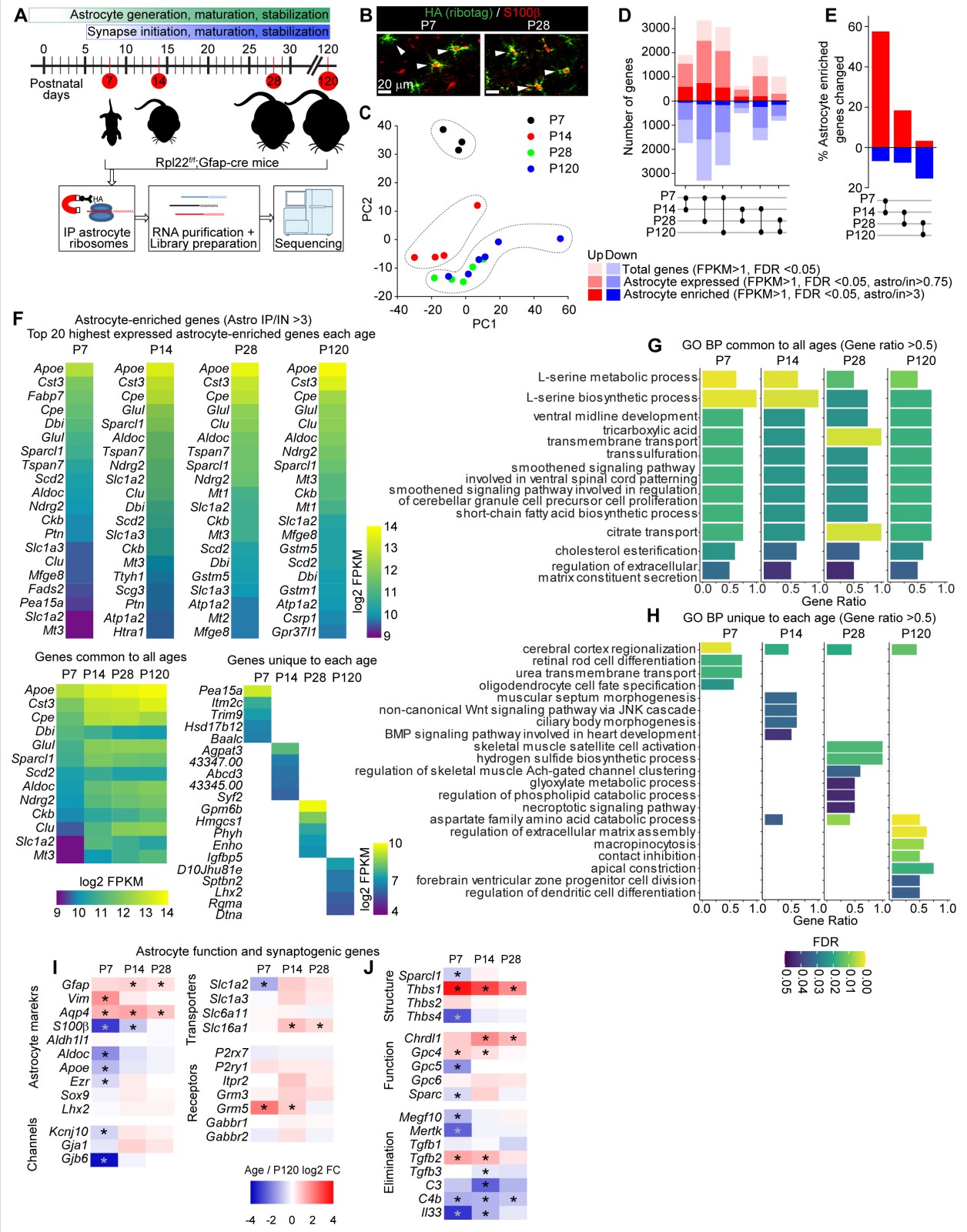

**Figure 1.** Developmental profiling of the astrocyte transcriptome in the postnatal visual cortex (VC). See also *Figure 1—figure supplement 1*, *Figure 1—source data 1*. (**A**) Outline of experiment: VCs from Rpl22-HA^f/+; Gfap-cre 73.12 mice were collected at different time points corresponding to synapse development and maturation, and subjected to Ribotag pulldown protocol, followed by RNA purification, library preparation and sequencing. (**B**) Example images of VC at postnatal day (P)7 and P28 as labeled, showing colocalization between Ribotag (green, HA) and astrocyte marker S100β

*Figure 1 continued on next page*

*Figure 1 continued*

(red). Scale bars = 20 µm. (**C**) Principal component analysis of RNAseq data shows P7 and P14 samples clustering separately from other ages, while P28 and P120 samples cluster together, suggesting similar gene expression profiles (N = 3 at P7, 4 at P14, 5 at P28, 3 at P120; for statistical comparisons, 3xP120 samples published in *Boisvert et al., 2018* were added to increase the power of the analysis, giving an N = 6 P120). (**D**) Pairwise comparison of differentially expressed genes (DEGs; red: upregulated; blue: downregulated) between each time point showing total genes (light hue, all DEGs identified with FPKM >1), astrocyte expressed genes (darker hue, expression level in pulldown sample/input >0.75), and astrocyte-enriched genes (dark hue, expression level in pulldown sample; astro/input >3). (**E**) Percent of all astrocyte-enriched genes that are differentially expressed between each age. Percent DEGs is highest between P7 and P14. (**F**) Heatmaps of top 20 astrocyte-enriched genes at each age, sorted by expression level, along with 13 genes common to all time points (left), and top 5 most enriched genes at each age (right). Colors represent log2 FPKM of expression level. (**G, H**) Gene Ontology (GO) terms analysis with String db of Biological Process (BP) in astrocyte-enriched genes at each time point. (**G**) Plot of GO terms common to all time points with a gene ratio >0.5. (**H**) Plot of GO terms unique to each age with a gene ratio >0.5. Bar length is gene ratio, fill color is false discovery rate (FDR). See also *Figure 1—source data 1B*. (**I, J**) Heatmaps of select genes related to astrocyte function (**I**) and synaptic regulation (**J**) plotted as log2 fold change (FC) at each age relative to P120. * Adjusted p value (FDR) < 0.05 by DESeq2 with Benjamini–Hochberg's correction when comparing P120 to each age.

The online version of this article includes the following figure supplement(s) for figure 1:

**Source data 1.** Temporal profiling of astrocyte transcriptome in the postnatal visual cortex (VC).

**Figure supplement 1.** Developmental profiling of the astrocyte transcriptome in the postnatal visual cortex (VC).

astrocytes globally. The expression of another well-known astrocytic gene *Aldh1l1* (*Morel et al., 2017*; *Figure 1I*) is stable across all ages, making it an optimal marker for astrocytes during both development and in the adult (*Srinivasan et al., 2016*). On the other hand, *S100b* expression is upregulated later in development, making it more suitable to mark adult astrocytes (*Figure 1I*). For genes that encode proteins important for astrocyte function, we found that the metabotropic glutamate receptor *Grm5* (mGluR5) is most highly expressed at P7 and then declines with maturation, confirming previous reports (*Catania et al., 1994*; *Sun et al., 2013*), while the glutamate transporter *Slc1a2* (Glt1) and the connexins (*Gja1*, *Gjb6*) are significantly upregulated from P14 onwards (*Figure 1I*).

## Astrocytic synapse-regulating genes show differential spatio-temporal expression patterns

Our goal is to characterize how astrocytes regulate synapses, so we next used the astrocyte Ribotag dataset to investigate the developmental expression changes of key synapse-regulating genes (*Figure 1J*). These include astrocyte-secreted thrombospondins (*Thbs*), which induce silent synapse formation. The family members expressed by astrocytes in the CNS show divergent expression, with *Thbs1* being significantly higher at P7 than later ages, whereas *Thbs4* is significantly higher at P120 than P7 (*Figure 1J*). This temporal expression profile fits with previous studies that have demonstrated important roles for *Thbs1* in initial synapse formation at P7 (*Christopherson et al., 2005*), and suggested roles for *Thbs4* in the adult brain (*Benner et al., 2013*). Similarly, glypican (*Gpc*) family members have a divergent expression. Though redundant in culture for their ability to regulate immature synapse formation (*Allen et al., 2012*), *Gpc4* and *Gpc6* show different temporal expression in vivo. While *Gpc4* is most highly expressed at P7 and gradually declines with maturation, *Gpc6* peaks at P14–P28. *Gpc5*, a glypican family member with yet unknown function, has low expression at P7, and is significantly increased at all later ages. Astrocyte-secreted chordin-like 1 (*Chrdl1*) regulates synapse maturation and its expression peaks at P14, confirming previous analysis (*Blanco-Suarez et al., 2018*; *Figure 1J*). These temporal expression profiles are not limited to factors that promote synapse formation. Astrocyte phagocytic receptors involved in synapse elimination, *Megf10* and *Mertk* (*Chung et al., 2013*), significantly increase in expression between P7 and P14, coincident with the initiation of synapse elimination.

Our RNA sequencing analysis demonstrates that in the developing VC synapse-regulating genes show differential temporal expression patterns. However, whether these levels are equal across astrocytes in all cortical layers throughout development is unknown. During the first postnatal weeks, when astrocytes regulate synapse development, they are still migrating and dividing (*Ge et al., 2012*). Yet, how developing astrocytes populate each of the cortical layers, and their ratio relative to other cortical cells across development has not been quantified. To address this, we utilized the well-established mouse line where astrocytes express GFP under the *Aldh1l1* promoter (*Cahoy et al., 2008*; *Dougherty et al., 2010*; *John Lin et al., 2017*; *Stogsdill et al., 2017*; *Tien et al., 2012*). Immunostaining of

brain sections from Aldh1l1-GFP mice at P7 and P28 with antibodies against known astrocyte markers ALDH1L1, S100β, and SOX9 showed high overlap between GFP and marker-positive cells, suggesting the majority of astrocytes in the VC express GFP in this mouse line (*Figure 2—figure supplement 1A–C*), further validating its usage. We quantified astrocyte numbers within each of the six neuronal layers at the developmental time points, which correspond to astrocyte and synapse development throughout the first postnatal month (P1, P4, P7, P14, P28; *Figure 2A–C*). At birth (P1), very few astrocytes are present, comprising 0.5–2% of the total cell number in the VC (represented as GFP-positive cells as a percentage of all cells marked by the nuclear dye DAPI), with a significantly higher percentage of astrocytes in deeper layers than upper layers (*Figure 2C and D*, *Figure 2—source data 1A*). The astrocyte percentage increases with development in all cortical layers, peaking at P21–P28. At this time, astrocytes are ~10% of total cell number in layers (L) 2–6, and ~50% in L1 (*Figure 2C and D*, *Figure 2—source data 1A*). As the brain develops, the distance between cells grows to accommodate the increase in cell size and complexity, as evident by a significant decrease in DAPI-positive nuclei per mm$^2$ of VC that occurs from P1 to P14–P28 (*Figure 2—figure supplement 1D*, *Figure 2—source data 1A*). Despite this decrease in total cell density, the proportion of astrocytes remains constant in all cortical layers and across ages, with the exception of L1–2/3, where it is significantly lower at P1 (*Figure 2C and E*). This stability in astrocyte density is likely explained by new astrocytes still being generated in the weeks after birth (*Ge et al., 2012*).

Having established the developmental profile of astrocytes, we next determined if there are layer-specific developmental changes in mRNA expression of synapse-regulating genes by performing single-molecule fluorescent in situ hybridization (smFISH; RNAscope) on brain sections of Aldh1l1-GFP mice. We probed for seven genes that regulate distinct aspects of synaptogenesis: active synapse-regulating – glypicans (*Gpc*) 4, 5, 6; synapse maturation regulating – chordin-like 1 (*Chrdl1*); and silent synapse-regulating – thrombospondins (*Thbs1, 2, 4*) (*Figure 2F–M*, *Figure 2—figure supplement 2*, *Figure 2—source data 1B and C*). Expression of each gene (represented as total area of thresholded signal in μm$^2$; labeled as thresh area) was analyzed within the territory of GFP-positive astrocytes in each cortical layer at similar time points as above: P4, P7, P14, and P28, when most alterations in astrocyte transcriptome (*Figure 1*) and synapse development occur (*Farhy-Tselnicker and Allen, 2018*), and excluding P1, when the numbers of astrocytes in the cortex are very low. A negative control probe was used to determine the minimal signal threshold of detection (*Figure 2—figure supplement 2H and I*).

We first analyzed glypicans, factors that promote active synapse formation. Bulk RNAseq (Ribotag) showed that *Gpc4* expression by astrocytes is highest at P7 and gradually declines with maturation (*Figure 1J*, *Figure 1—source data 1A*). Layer-specific analysis, however, shows that these differences are driven by astrocytes in L1: *Gpc4* expression decreases between P7 and P14 only in L1 astrocytes, staying stable across development in all other layers (thresh area [μm$^2$] L1: P4 3.1 ± 0.5; P7 4.1 ± 0.1; P14 1.7 ± 0.2; P28 1.1 ± 0.3; *Figure 2F and G*, *Figure 2—source data 1B and C*). *Gpc6*, on the other hand, peaks at P14–P28 in bulk sequencing. Layer-specific analysis showed that this increase occurs in astrocytes between P7 and P14 in L2–5, with significant upregulation in L5, and remaining high at P28 (thresh area [μm$^2$] L5: P4 2.5 ± 0.4; P7 2.5 ± 0.4; P14 4.4 ± 0.4; P28 3.1 ± 0.4; *Figure 2H and I*, *Figure 2—source data 1B and C*). *Gpc5* is strongly upregulated at P14 and remains high at P28 in astrocytes in all layers, matching the bulk RNAseq data (thresh area [μm$^2$] L1: P4 0.5 ± 0.1; P7 1.3 ± 0.2; P14 5.9 ± 0.7; P28 5.3 ± 1.1; *Figure 2J and K*, *Figure 2—source data 1B and C*).

The expression of the synapse maturation factor *Chrdl1* peaks at P14 in the bulk RNAseq data (*Figure 1J*, *Figure 1—source data 1A*). Spatial analysis revealed that this increase from P7 to P14 is layer-specific, with the largest increase in *Chrdl1* occurring in upper layer astrocytes with significant upregulation in L2/3 (thresh area [μm$^2$] L2/3: P4 4.1 ± 0.2; P7 4.1 ± 1; P14 9.9 ± 0.6; P28 7.6 ± 1; *Figure 2L and M*, *Figure 2—source data 1B and C*). Further, at its peak of expression, *Chrdl1* is highest in L1–4 astrocytes compared to L5–6, demonstrating a heterogeneous expression across layers, as previously reported (*Bayraktar et al., 2020*; *Blanco-Suarez et al., 2018*). Finally, we analyzed astrocyte thrombospondins, factors that induce silent synapse formation (*Figure 2—figure supplement 2B-G*). Thbs mRNA levels in the VC are much lower at their peak expression than glypicans or *Chrdl1*, consistent with our bulk RNAseq results (*Figure 1J*, *Figure 1—source data 1A*), and previous studies showing low thrombospondin expression in the resting state, and an upregulation by learning or injury (*Nagai et al., 2019*; *Tyzack et al., 2014*). Nevertheless, we observed an increase in *Thbs2*

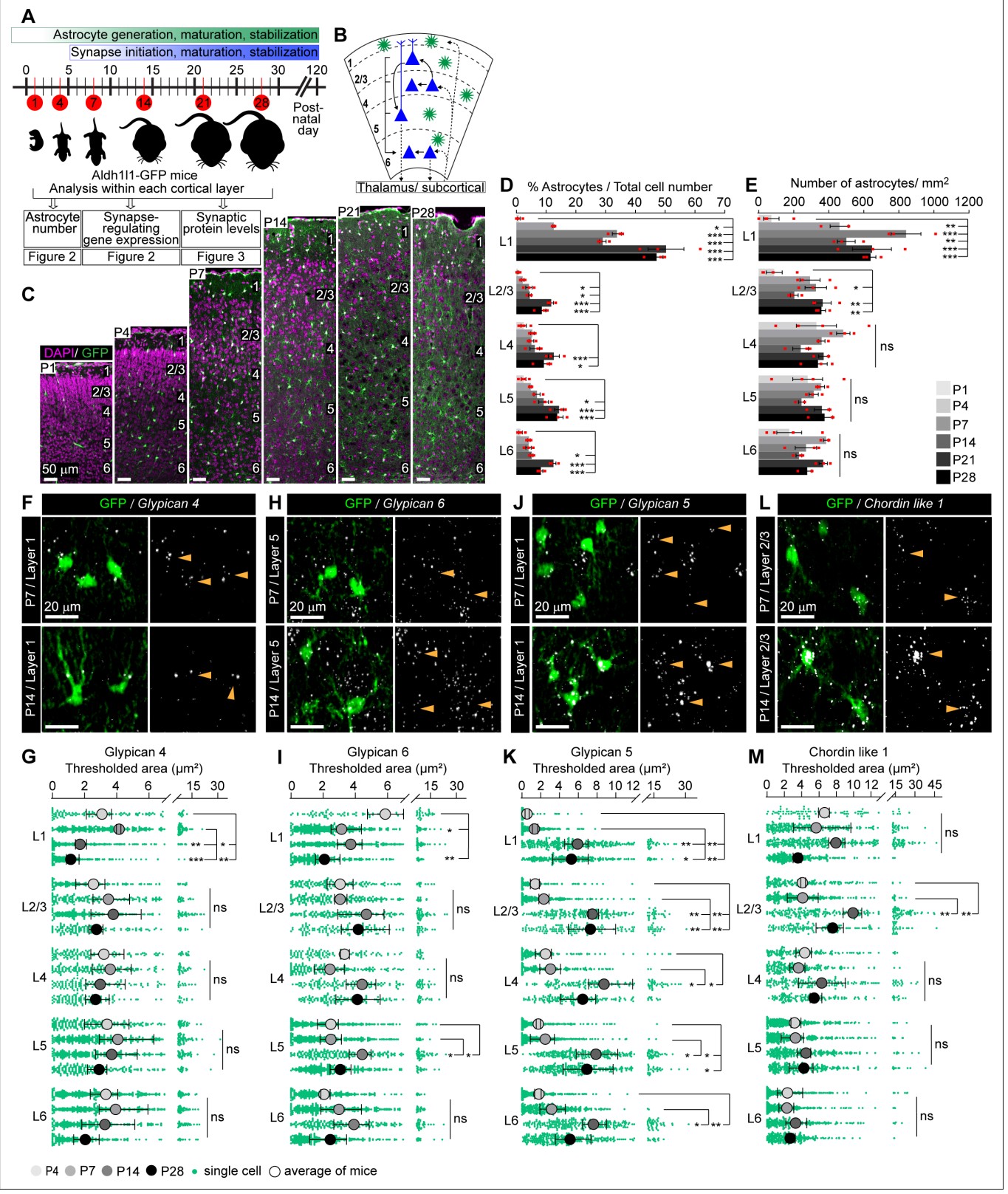

**Figure 2.** Astrocytic synapse-regulating genes show differential spatio-temporal expression patterns. See also *Figure 2—figure supplements 1 and 2*, *Figure 2—source data 1*. (**A**) Schematic of experiments described in *Figures 2–3*: visual cortex (VC) sections from Aldh1l1-GFP mice analyzed for astrocyte numbers, gene expression, and synaptic proteins at developmental time points as indicated. (**B**) Diagram of VC depicting neuronal (blue) laminar arrangement and connectivity (arrows). Astrocytes (green) are present in all layers. (**C–E**) Astrocyte number increases in the VC across

*Figure 2 continued on next page*

*Figure 2 continued*

development. (**C**). Example images of the VC from Aldh1l1-GFP mice at time points analyzed. GFP marks astrocytes (green), DAPI (magenta) labels nuclei. Layers (L) labeled by numbers on the right in each panel. (**D**) Quantification of (**C**), astrocytes as a percentage of total cells within each cortical layer. (**E**) Quantification of (**C**), number of astrocytes per mm$^2$ of VC within each layer. Scale bar in (**C**): 50 µm; N = 4 mice for postnatal day (P)1; N = 3 mice for P4–P28. Graphs show mean ± s.e.m., red squares are average of individual mouse. *p≤0.05, **p<0.01, ***p<0.001, ns (not significant) by one-way ANOVA comparing expression between time points within each layer; see also *Figure 2—source data 1A*. (**F–M**) Synapse-regulating genes show differential spatio-temporal expression patterns. (**F, H, J, L**) Example images showing *Gpc4*, *Gpc6*, *Gpc5*, or *Chrdl1* mRNA (white) in astrocytes (green) at each age and layer as labeled. Merged panel on the left, single-channel probe panel on the right. (**G, I, K, M**) Quantification of (**F, H, J, L**), respectively. (**F**) *Gpc4* expression is reduced at P14 specifically in L1. (**H**) *Gpc6* expression is increased at P14 in L5. (**J**) *Gpc5* expression is increased at P14 in all layers. (**L**) *Chrdl1* expression is increased at P14 in L2/3. Scale bars in (**F, H, J, L**) = 20 µm. Arrowheads in single-channel panel mark astrocyte cells on the left. N = 3 mice/age for each probe. Data presented as scatter with mean + range. Green dots are mRNA signal measured in individual astrocytes. Large circles colored according to age are average signal. N = 3 mice/age, n = ~50–350 astrocytes/per age total; averages and statistical analysis are calculated based on N = 3, i.e., data per mouse. *p≤0.05, **p<0.01, ***p<0.001, ns (not significant) by one-way ANOVA comparing expression between time points within each layer. See also *Figure 2—source data 1B and C*.

The online version of this article includes the following source data and figure supplement(s) for figure 2:

**Source data 1.** Astrocytic synapse-regulating genes show differential spatio-temporal expression patterns.

**Figure supplement 1.** Astrocytic synapse-regulating genes show differential spatio-temporal expression patterns.

**Figure supplement 2.** Astrocytic synapse-regulating genes show differential spatio-temporal expression patterns.

expression in L4–5 at P28 (thresh area [µm$^2$] L5: P4 0.7 ± 0.1; P7 0.7 ± 0.19; P14 1 ± 0.1; P28 1.8 ± 0.3; *Figure 2—figure supplement 2D and E*, *Figure 2—source data 1B*); and an increase in *Thbs4* at P28 in all layers (thresh area [µm$^2$] L1: P4 0.1 ± 0.01; P7 0.1 ± 0.1; P14 0.3 ± 0.1; P28 1.9 ± 0.5; *Figure 2—figure supplement 2F and G*, *Figure 2—source data 1*).

The seven astrocyte synapse-regulating genes we analyzed using smFISH all regulate formation and function of excitatory glutamatergic synapses. In the case of *Gpc4*, *Gpc6*, and *Chrdl1*, they regulate glutamatergic synapse development by primarily affecting localization of AMPA glutamate receptor (AMPAR) subunits (*Allen et al., 2012*; *Blanco-Suarez et al., 2018*). Previous studies in the rat somatosensory cortex have identified developmental alterations in AMPAR subunit expression (*Brill and Huguenard, 2008*; *Kumar et al., 2002*); however, their spatio-temporal expression patterns in the developing mouse VC, and whether these correlate with the expression of astrocytic genes that regulate them, have not been systematically analyzed (*Figure 3A and B*). To detect postsynaptic AMPA glutamate receptors, we stained for GLUA1 and GLUA2 subunits (*Figure 3C–F*, *Figure 3—source data 1*). To detect the corresponding glutamatergic presynaptic terminals, we stained for VGLUT1 to identify local cortico-cortical connections (*Figure 3G and H*, *Figure 3—figure supplement 1A*, *Figure 3—source data 1*) and VGLUT2 to identify thalamocortical connections (*Figure 3I and J*, *Figure 3—figure supplement 1B*, *Figure 3—source data 1*; *Fremeau et al., 2001*). As expected based on the literature, GLUA1 levels peak at earlier time points than GLUA2 (*Figure 3C and D*, *Figure 3—source data 1*). GLUA2 immunoreactivity significantly increases from P7 to P14 in L1–5, and then remains stable to P28 (*Figure 3E and F*, *Figure 3—source data 1*). At all ages, the levels of GLUA1 and GLUA2 are significantly higher in L1 than all other layers, consistent with L1 being rich in synaptic connections (*Figure 3—source data 1*; *Douglas and Martin, 2004*). VGLUT1 immunoreactivity greatly increases between P7 and P14 in all cortical layers and remains stable at later ages (*Figure 3G and H*, *Figure 3—figure supplement 1A*, *Figure 3—source data 1*). VGLUT2 levels steadily increase from P1 to P14, and then remain stable. VGLUT2 immunoreactivity is significantly higher in L1 and L4 than other layers at all ages, consistent with these being thalamic innervation zones and previous findings in rat cortex (*Figure 3I and J*, *Figure 3—figure supplement 1B*, *Figure 3—source data 1*; *Boulland et al., 2004*; *Conti et al., 2005*; *Fremeau et al., 2001*; *Lopez-Bendito and Molnar, 2003*). Taken together, these data show that synaptic proteins show complex spatio-temporal expression patterns, with multiple significant changes occurring between P7 and P14, as also observed for astrocyte gene expression.

In summary, the transcriptomic analysis reveals significant changes in astrocyte gene expression across development, with the most prominent changes occurring between P7 and later ages, a time between synapse initiation and maturation. We have further determined the spatio-temporal expression profile of key astrocyte synapse-regulating factors, identifying divergent developmental and layer-specific expression patterns within the same families of genes. These findings strongly suggest

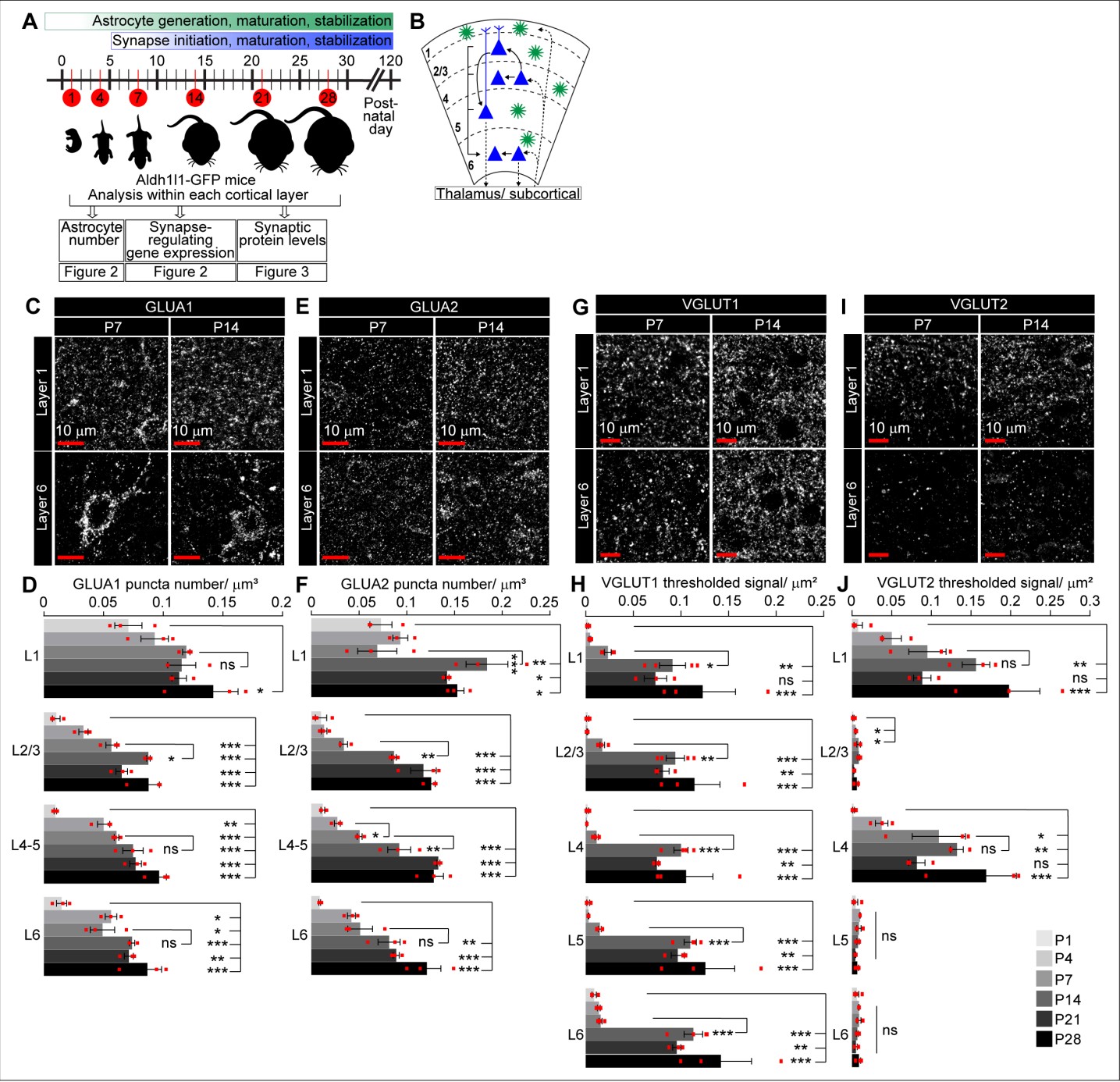

**Figure 3.** Spatio-temporal profiling of synaptic protein expression in the postnatal visual cortex (VC). See also *Figure 3—figure supplement 1*, *Figure 3—source data 1*. (**A**) Schematic of experiments described in *Figures 2–3*: VC sections from Aldh1l1-GFP mice analyzed for astrocyte numbers, gene expression, and synaptic proteins at developmental time points as indicated. (**B**) Diagram of VC depicting neuronal (blue) laminar arrangement and connectivity (arrows). Astrocytes (green) are present in all layers. (**C–F**) Developmental expression pattern of the postsynaptic AMPARs GLUA1 and GLUA2 subunits within each cortical layer. (**C, E**) Example images of GLUA1 or GLUA2 protein levels (white puncta). (**D, F**) Quantification of (**C, E**) number of GLUA1 or GLUA2 puncta per cortical volume ($\mu m^3$). GLUA1 expression is increased from postnatal day (P)1 to P28 in all layers. GLUA2 expression is increased from P1 to P28 in all layers, and between P7 and P14 in L1–5. (**G–J**) Developmental expression pattern of the presynaptic proteins VGLUT1 and VGLUT2 in each cortical layer. (**G, I**) Example images of VGLUT1 or VGLUT2 protein levels (white puncta). (**H, J**) Quantification of (**G, I**) density of VGLUT1 or VGLUT2 signal as total area of thresholded signal per $\mu m^2$. VGLUT1 expression increases in all layers between P7 and P14. VGLUT2 expression increases steadily from P1 to P28 in L1 and L4. Scale bars = 10 $\mu m$. N = 3 mice/age. Graphs show mean ± s.e.m., red squares average of individual mouse. *p≤0.05, **p<0.01, ***p<0.001, ns (not significant) p>0.05 by one-way ANOVA comparing expression between time points

*Figure 3 continued on next page*

*Figure 3 continued*

within each layer.

The online version of this article includes the following figure supplement(s) for figure 3:

**Source data 1.** Spatio-temporal profiling of synaptic protein levels.

**Figure supplement 1.** Spatio-temporal profiling of synaptic protein expression in the postnatal visual cortex (VC).

that astrocyte expression of synapse-regulating genes is closely tied to the developmental stage of the cortex, which features both changes in neuronal and astrocyte activities across development. In addition to revealing important information about the developmental changes in expression of synapse-regulating genes, these transcriptomic data can be further utilized to inform further studies on astrocyte development. The complete dataset and GO term list are available in *Figure 1—source data 1*.

## Thalamic glutamate release tunes astrocyte expression of synapse-regulating genes

Having found broad differences in astrocyte expression of synapse-regulating genes across cortical layers, we next asked what are the possible physiological mechanisms that regulate these layer-specific alterations between P7 and P14 in the developing VC in vivo. Our experiments using cultured astrocytes and neurons show that GPC4 protein secretion from astrocytes is significantly reduced in the presence of neurons (*Figure 4—figure supplement 1A*, *Figure 4—figure supplement 1—source data 1*) or when astrocytes are incubated with neurotransmitters including glutamate, adenosine, or ATP (*Figure 4—figure supplement 1B*, *Figure 4—figure supplement 1—source data 1*). Others have reported a similar effect on *Gpc4* mRNA expression (*Hasel et al., 2017*), suggesting that neuronal activity can influence expression and release of synapse-regulating factors from astrocytes. Indeed, significant alterations in the activity patterns of both glutamatergic and GABAergic neurons occurs in the VC at around P14 (*Espinosa and Stryker, 2012*). Since the developmental expression changes in *Gpc4* and *Chrdl1* occurred mostly in the upper cortical layers innervated by thalamic neurons, we hypothesized that levels of these genes may be regulated by changes in the activity of thalamic neurons that occur upon eye opening at ~P12, when there is a transition from spontaneous to visually evoked activity in the retina that is relayed via the thalamus to the VC.

To investigate this, we generated mice where the release of glutamate from thalamocortical terminals in the VC is perturbed by knocking out the vesicular glutamate transporter VGlut2 (*Slc17a6*) from neurons in the dLGN. Knockout of VGlut2 has been previously shown to abolish presynaptic release of glutamate in VGlut2-expressing neurons, in full or conditional knockout mouse models (*Wallén-Mackenzie et al., 2010*). We crossed *Slc17a6*^f/f mice (Slc17a6^tm1Lowl/J; labeled as VGlut2 WT) to an RORα cre line (Rora^tm1(cre)Ddmo; *Slc17a6*^f/f;cre labeled as VGlut2 cKO), where cre recombinase is expressed in thalamic neurons including the dLGN (*Figure 4A*; *Chou et al., 2013*; *Farhy-Tselnicker et al., 2017*). Immunostaining experiments showed a significant decrease in VGLUT2 signal in the VC of VGlut2 cKO mice compared to WT at P14 (*Figure 4B*). Notably, VGlut2 cKO did not strongly affect the levels of VGLUT1, which marks cortico-cortical connections (*Figure 4C*; *Wallén-Mackenzie et al., 2010*), though a small but significant increase in VGLUT1 immunoreactivity was observed in L4, a major target of thalamocortical projections. This suggests that the normal upregulation in VGLUT1 that occurs at P14 across all other cortical layers (*Figure 3*) is either intrinsic to the cortical neurons and/or regulated by mechanisms other than dLGN-VC-evoked activity.

A lack of VGLUT2 immunoreactivity in the VC could also result from an absence of thalamic axons innervating their target regions. To test whether that is the case, we crossed RORα cre and VGlut2 cKO mice with a tdTomato reporter line (B6.C-Gt(ROSA)26Sor^tm14(CAG-tdTomato)Hze/J; labeled as tdTomato) to visualize dLGN axons (*Figure 4A*). All cre+ tdTomato+ groups (VGlut2 WT (*Slc17a6*^+/+), VGlut2 cHet (*Slc17a6*^f/+), and VGlut2 cKO (*Slc17a6*^f/f)) showed a comparable number and volume of tdTomato-labeled projections in L1 and L4 of the VC (*Figure 4—figure supplement 2A and B*). Analysis of VGLUT2 puncta colocalized with tdTomato-positive axons showed a significant decrease in number in VGlut2 cHet and VGlut2 cKO compared to WT (*Figure 4—figure supplement 2A, C*). These results show that in the VGlut2 cKO mice thalamic axons are present at their target layers in the VC but lack

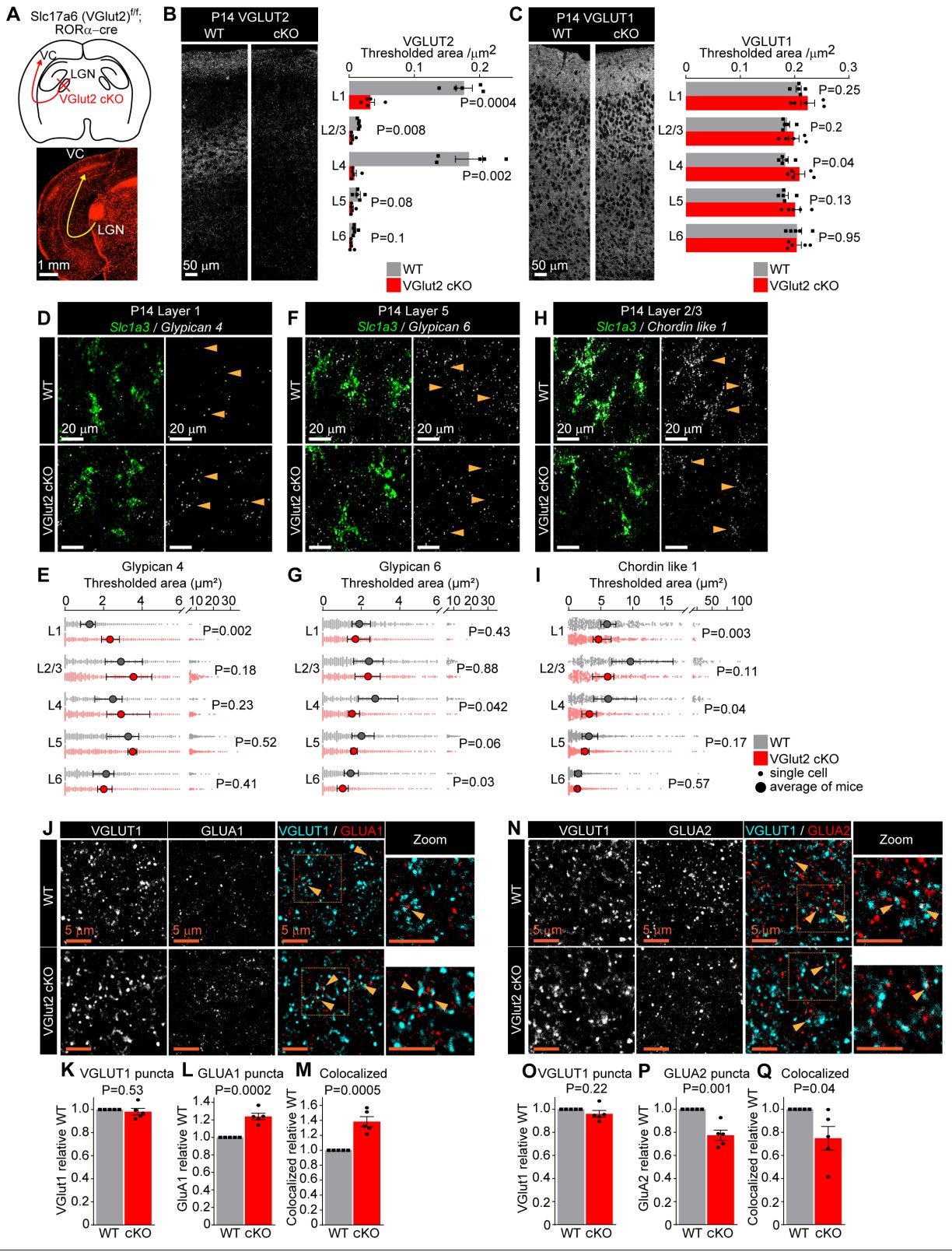

**Figure 4.** Thalamic glutamate release tunes astrocyte expression of synapse-regulating genes. See also *Figure 4—figure supplements 1 and 2*, *Figure 4—source data 1*. (**A**) Schematic of the experiment: VGLUT2 is removed from presynaptic terminals of neurons in the lateral geniculate nucleus of the thalamus (LGN) that project to the visual cortex (VC) by crossing *Slc17a6* [f/f] mouse (WT) with RORαcre mouse line (VGlut2 cKO). Bottom: image of tdTomato reporter expression in the LGN and the VC, when RORαcre mouse is crossed with cre-dependent tdTomato reporter mouse (Ai14). (**B**)

*Figure 4 continued on next page*

*Figure 4 continued*

VGLUT2 level in the VC is significantly reduced in VGlut2 cKO mice. Example images of VGLUT2 immunostaining in each genotype and quantification of the thresholded signal within each cortical layer. (**C**) VGLUT1 level is unaltered in VGlut2 cKO mice. Example images of VGLUT1 immunostaining and quantification. In (**B, C**), plots show mean signal ± s.e.m. Squares and circles above each bar are the average of signal in each mouse. N = 5 mice/genotype. Scale bar = 50 µm. Statistical analysis by t-test within each layer. p-Value on each plot. (**D–I**) mRNA expression of astrocyte synapse-regulating genes is altered in VGlut2 cKO at postnatal day (P)14. (**D, F, H**) Example images of in situ hybridization of *Gpc4*, *Gpc6*, and *Chrdl1* mRNA (white) as labeled; astrocyte marker *Slc1a3* (Glast, green). Merged panel on the left, single-channel probe panel on the right. (**E, G, I**) Quantification of (**D, F, H**), respectively. (**E**) *Gpc4* mRNA expression is increased in L1; (**G**) *Gpc6* mRNA expression is decreased in L4–6. (**I**) *Chrdl1* mRNA expression is decreased in L1–4 in VGlut2 cKO mice. Data presented as scatter with mean + range. Gray or red dots are mRNA signals measured in individual astrocyte in WT and VGlut2 cKO, respectively. Large circles are the average signal. N = 5 mice/genotype, n = ~200–450 astrocytes/per age total (average and statistical analysis are calculated based on N = 5, i.e., per mouse). Arrowheads in single-channel panel mark astrocytes. Scale bar = 20 µm. See also *Figure 4—source data 1*. Statistical analysis by t-test within each layer. p-Value on each plot. (**J–M**) Increase in GLUA1 protein levels and colocalization between GLUA1 and VGLUT1 in L1 of the VC in VGlut2 cKO mice compared to WT at P14. (**J**) Example images from WT (top) and cKO (bottom), VGLUT1 in cyan and GLUA1 in red. (**K–M**) Quantification of VGLUT1, GLUA1, and colocalized puncta, respectively, normalized to WT. (**N–Q**) Decrease in GLUA2 protein levels and colocalization between GLUA2 and VGLUT1 in L1 of the VC in VGlut2 cKO mice compared to WT at P14. (**N**) Example images from WT (top) and cKO (bottom), VGLUT1 in cyan and GLUA2 in red. (**O–Q**) Quantification of VGLUT1, GLUA2, and colocalized puncta, respectively, normalized to WT. In (**K–M**) and (**O–Q**), data presented as mean ± s.e.m., squares and circles above each bar are mean fold change of each mouse. N = 5 mice/genotype. Arrowheads mark representative colocalized puncta in (**J, N**). Inset panels on the right show enlarged colocalized image from box in (**J, N**). Scale bar = 5 µm. In (**J–Q**), WT is *Slc17a6* $^{+/+; ROR\alpha cre+;tdTomato+}$; cKO is *Slc17a6* $^{f/f;ROR\alpha cre+;tdTomato+}$. Statistical analysis by t-test, p-value on each plot.

The online version of this article includes the following figure supplement(s) for figure 4:

**Source data 1.** Neuronal activity regulates astrocytic expression of synapse-regulating genes.

**Figure supplement 1.** Thalamic glutamate release tunes astrocyte expression of synapse-regulating genes.

**Figure supplement 1—source data 1.** Neuronal activity regulates astrocytic expression of synapse-regulating genes.

**Figure supplement 2.** Thalamic glutamate release tunes astrocyte expression of synapse-regulating genes.

VGLUT2, as has been shown in studies performing similar manipulations, suggesting that they are functionally silent (*Li et al., 2013*; *Zechel et al., 2016*).

Does the lack of thalamocortical glutamate release influence the expression of synapse-regulating genes in astrocytes? To address this, we performed smFISH probing for *Gpc4*, *Gpc6*, or *Chrdl1*, along with a probe for the glutamate transporter *Slc1a3* (Glast) (*Ullensvang et al., 1997*) to label astrocytes (*Figure 4D–I*, *Figure 4—figure supplement 2D–J*, *Figure 4—source data 1*). The number of cortical astrocytes marked by *Slc1a3*, and the expression level of *Slc1a3* mRNA, is not affected by VGlut2 cKO in any cortical layer at P14, showing that gross astrocyte development proceeds normally in the absence of thalamic input (*Figure 4—figure supplement 2D and E*). During normal development, *Gpc4* expression significantly decreases between P7 and P14 specifically in L1 astrocytes (*Figure 2F and G*). In VGlut2 cKO mice, this change no longer occurs (*Figure 4D and E*). *Gpc4* expression is significantly increased in VGlut2 cKO compared to WT at P14 specifically in L1 astrocytes and unchanged in all other layers (thresh area [µm²]: L1 WT 1.3 ± 0.1; cKO 2.3 ± 0.2; *Figure 4D and E*, *Figure 4—source data 1*). *Gpc6* expression is normally increasing in astrocytes in deep layers between P7 and P14 (*Figure 2H and I*). In the absence of thalamic glutamate release, *Gpc6* is lower in astrocytes in layers 4 and 5 than in the WT at P14, showing that the normal developmental upregulation has been blocked (thresh area [µm²]: L4 WT 2.7 ± 0.4; cKO 1.5 ± 0.1; *Figure 4F and G*, *Figure 4—source data 1*). Similarly, *Chrdl1* expression normally increases in upper layer astrocytes between P7 and P14 (*Figure 2L and M*); however, it is significantly decreased in VGlut2 cKO compared to WT specifically in astrocytes in upper layers (1–4), and not affected in deep layers at P14 (thresh area [µm²]: L1 WT 6 ± 0.4; cKO 4.6 ± 0.5; *Figure 4H and I*, *Figure 4—source data 1*). To determine if these alterations are due to the loss of thalamic neuron glutamate release at a specific developmental time (e.g., eye opening at ~P12), we analyzed the expression of *Gpc4*, *Gpc6*, and *Chrdl1* at P7 (prior to eye opening). We found no difference in expression of any of these genes (*Figure 4—figure supplement 2H–J*), nor any difference in the number of cortical astrocytes marked by *Slc1a3*, or the expression level of *Slc1a3* mRNA (*Figure 4—figure supplement 2F and G*). These results show that during VC development from P7 to P14 glutamate release from thalamic neurons regulates the developmental expression of astrocyte *Gpc4*, *Gpc6*, and *Chrdl1* in a layer-specific manner.

Are there any consequences of decreased glutamate release and subsequent altered expression of astrocyte synapse-regulating genes on synaptic development? To address this, we performed

immunostaining for pre- and postsynaptic proteins in VGlut2 cKO mice and their WT controls in L1 of VC at P14. We first analyzed cortico-cortical synapses marked by VGLUT1 and found that, as in the low-resolution characterization (*Figure 4C*), there is no change in VGLUT1 puncta in the absence of VGLUT2 (*Figure 4J, K, N and O*). In the case of GLUA1 containing AMPARs, which are regulated by *Gpc4*, we found a significant increase in the number of GLUA1 puncta and their colocalization with VGLUT1 in the VGlut2 cKO, correlating with the observed increase in *Gpc4* (GLUA1 FC 1.24 ± 0.04; *Figure 4J and L*; Coloc FC 1.38 ± 0.07; *Figure 4J and M*). For GLUA2 containing AMPARs, which are regulated by *Chrdl1*, we found a significant decrease in both the number of GLUA2 puncta and the number of colocalized GLUA2-VGLUT1 puncta in VGlut2 cKO mice compared to WT, correlating with the observed decrease in *Chrdl1* (GLUA2 FC 0.77 ± 0.04; *Figure 4N and P*; Coloc FC 0.75 ± 0.1; *Figure 4N and Q*). We asked if similar effects are also present at thalamocortical synapses. Since VGLUT2 is absent in cKO mice, we used Slc17a6$^{f/f; ROR\alpha cre;tdTomato}$ (labeled as VGlut2 cKO; tdTomato) to label thalamic axons (*Figure 4A*, *Figure 4—figure supplement 2A*) and identified presynaptic active zones within tdTomato axons by immunostaining for the presynaptic marker Bassoon (*Figure 4—figure supplement 2K and O*). We found no difference in the number of Bassoon puncta colocalized with tdTomato between the WT and cKO mice, a further indication that synapses form in the absence of VGLUT2 (*Figure 4—figure supplement 2K, L, O, P*), and fitting with findings from mice that globally lack presynaptic release (*Verhage et al., 2000*). As is the case for cortico-cortical synapses, we found an increase in GLUA1 and colocalization of GLUA1 with Bassoon and tdTomato (GLUA1 FC 1.21 ± 0.09; *Figure 4—figure supplement 2K and M*; Coloc FC 1.36 ± 0.23; *Figure 4—figure supplement 2K and N*), and a decrease in total GLUA2 and GLUA2-Bassoon synapses (GLUA2 FC 0.70 ± 0.08; *Figure 4—figure supplement 2O and Q*; Coloc FC 0.65 ± 0.13; *Figure 4—figure supplement 2O and R*).

These results show that synaptic GLUA1 and GLUA2 levels are altered in VGlut2 cKO VC in the direction which follows the change in astrocytic expression of *Gpc4* (which recruits GLUA1) and *Chrdl1* (which recruits GLUA2) in L1. These correlated changes in astrocyte genes and synaptic proteins suggest a disruption in synapse maturation in the VC at P14 in the absence of thalamocortical glutamate release that may be mediated by astrocytes.

## Astrocytic calcium signaling tunes expression of synapse-regulating genes

Since we observed that changes in thalamic glutamate release influence the expression of astrocyte synapse-regulating genes, we next asked how perturbing the astrocyte response to neurotransmitters affects the expression of *Gpc4*, *Gpc6,* and *Chrdl1*. Astrocytes express many neurotransmitter receptors, in particular GPCRs, and respond to neurotransmitters with increased intracellular calcium (*Kofuji and Araque, 2021*; *Porter and McCarthy, 1997*). In the case of somal increases in calcium, which have the potential to regulate expression of activity-regulated genes, most of this increase is mediated by the release of calcium from intracellular stores via IP3R2 (*Itpr2*) (*Srinivasan et al., 2015*). We therefore asked if blunting astrocyte calcium signaling by removing store-mediated calcium release using *Itpr2* KO mice (Itpr2$^{tm1.1Chen}$; labeled Ip3r2 KO; *Figure 5A and B*) has an impact on the expression of synapse-regulating genes.

To determine this, we performed smFISH on the VC of P14 Ip3r2 KO and WT mice, marking astrocytes with a probe against *Slc1a3* along with *Gpc4*, *Gpc6*, or *Chrdl1*. At P14, knocking out *Ip3r2* does not affect the number of astrocytes or the expression levels of *Slc1a3*, showing astrocytes develop grossly normally when store-mediated calcium release is diminished (*Figure 5—figure supplement 1A* and B, *Figure 5—source data 1*, *Figure 5—source data 2*). However, loss of IP3R2 does affect expression of synapse-regulating genes. In the case of *Gpc4*, the mRNA level is reduced in astrocytes in all layers, with a significant decrease occurring in L1, 4, and 6 (thresh area [μm$^2$]: L1 WT 1.5 ± 0.1; KO 1.1 ± 0.1; *Figure 5C and D*, *Figure 5—source data 1*). For *Gpc6*, there is no difference in the mRNA level between Ip3r2 KO and WT in astrocytes in any layer (*Figure 5E and F*, *Figure 5—source data 1*). *Chrdl1* expression is increased in astrocytes in all layers, with a significant increase occurring in L1, 2/3, and 5 (thresh area [μm$^2$]: L1 WT 7.3 ± 1.1; KO 10.6 ± 1; *Figure 5G and H*, *Figure 5—source data 1*). To ask if these alterations are present throughout development, we performed the same analysis at P7. As with P14, at P7 there is no change in astrocyte number or *Slc1a3* mRNA signal, showing astrocytes develop grossly normally (*Figure 5—figure supplement*

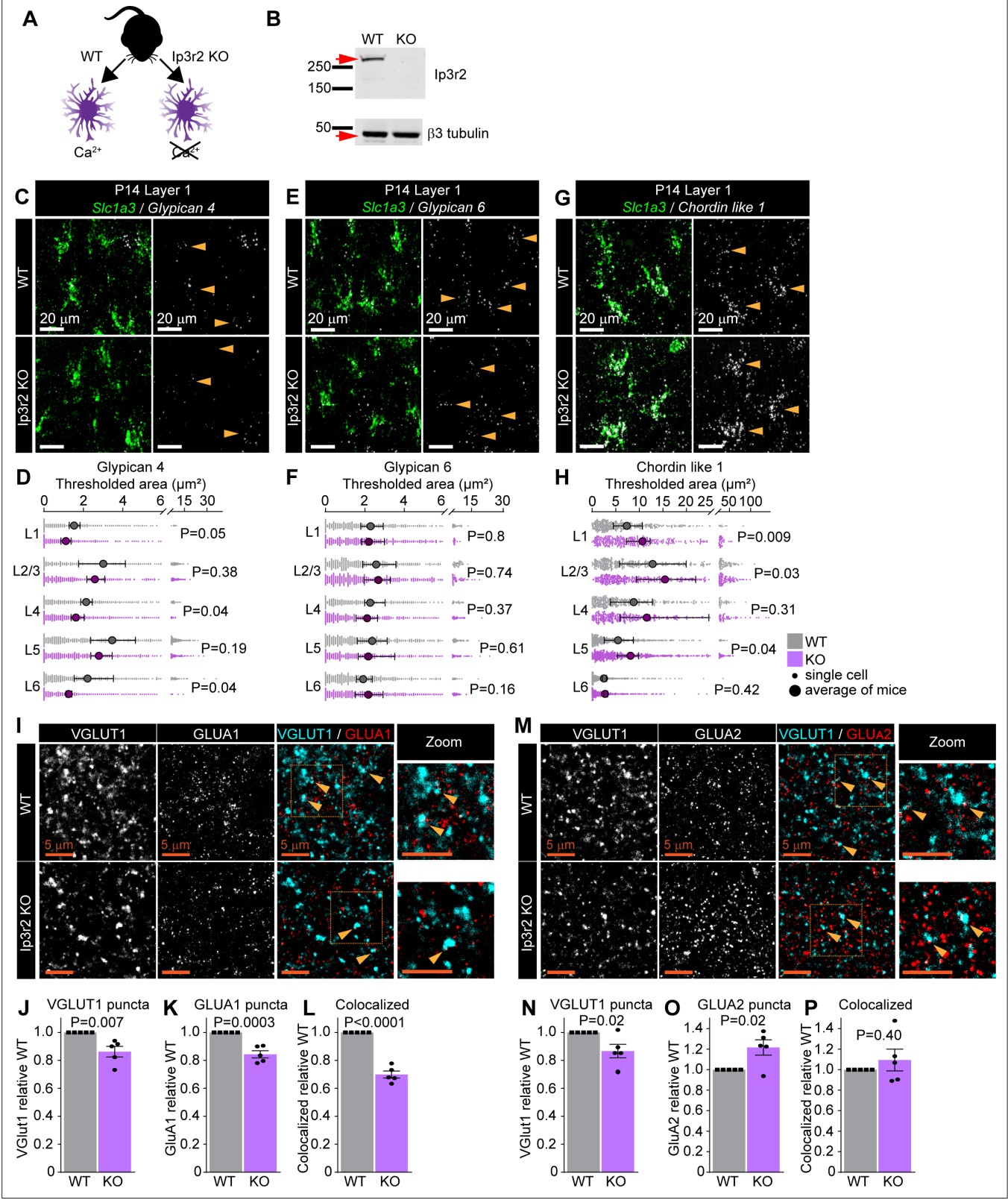

**Figure 5.** Astrocytic calcium signaling tunes expression of synapse-regulating genes. See also *Figure 5—figure supplement 1*, *Figure 5—source data 1* and *2*. (**A**) Schematic of comparison. Lack of the IP3R2 receptor results in diminished Ca²⁺ transients in astrocytes. (**B**) Validation of *Itpr2* KO model. Western blot image shows absence of IP3R2 signal in visual cortex (VC) of KO mice. (**C–H**) mRNA expression of astrocyte synapse-regulating genes is altered in Ip3r2 KO mice at postnatal day (P)14. (**C, E, G**) Example images of in situ hybridization of *Gpc4*, *Gpc6*, and *Chrdl1* mRNA (white) as

*Figure 5 continued on next page*

Figure 5 continued

labeled; astrocyte marker *Slc1a3* (Glast, green). Merged panel on the left, single-channel probe panel on the right. (**D, F, H**) Quantification of (**C, E, G**), respectively. (**D**) *Gpc4* mRNA expression is decreased in several layers of the VC in Ip3r2 KO mice. (**F**) *Gpc6* mRNA expression is unaltered in the VC in Ip3r2 KO mice. (**H**) *Chrdl1* mRNA expression is increased in several layers of the VC in Ip3r2 KO mice. Data presented as scatter with mean + range. Gray or purple dots are mRNA signals measured in individual astrocyte in WT and Ip3r2 KO, respectively. Large circles colored according to genotype are average signal. N = 5 mice/genotype, n = ~200–450 astrocytes/per age total (average and statistical analysis are calculated based on N = 5, i.e., per mouse). Arrowheads in single-channel panel mark astrocytes. Scale bar = 20 μm. Statistical analysis by t-test within each layer. p value on each plot. (**I–L**) Decrease in VGLUT1, GLUA1 protein levels, and colocalization between GLUA1 and VGLUT1 in L1 of the VC in Ip3r2 KO mice at P14. (**I**) Example images from WT (top) and KO (bottom), VGLUT1 in cyan and GLUA1 in red. (**J–L**) Quantification of VGLUT1, GLUA1, and colocalized puncta, respectively, normalized to WT. (**M–P**) Decrease in VGLUT1, and increase GLUA2 protein levels, with no change in colocalization between GLUA2 and VGLUT1 in L1 of the VC in Ip3r2 KO mice at P14. (**M**) Example images from WT (top) and KO (bottom), VGLUT1 in cyan, and GLUA2 in red. (**N–P**) Quantification of VGLUT1, GLUA2, and colocalized puncta, respectively, normalized to WT. In (**J–L**) and (**N–P**), data presented as mean ± s.e.m. squares and circles above each bar are mean fold change of each mouse. N = 5 mice/genotype. Arrowheads mark representative colocalized puncta. Inset panels on the right show enlarged colocalized image from box in (**M**). Scale bar = 5 μm. Statistical analysis by t-test, p-value on each plot.

The online version of this article includes the following figure supplement(s) for figure 5:

**Source data 1.** Astrocyte calcium activity regulates astrocytic expression of synapse-regulating genes.

**Source data 2.** Astrocyte calcium activity regulates astrocytic expression of synapse-regulating genes.

**Figure supplement 1.** Astrocytic calcium signaling tunes expression of synapse-regulating genes.

*1C and D*). In the case of *Gpc4* and *Chrdl1,* there is no difference in expression between Ip3r2 KO and WT at P7 (***Figure 5—figure supplement 1E and G***), whereas for *Gpc6* there is a significant increase in the Ip3r2 KO restricted to L4 (***Figure 5—figure supplement 1F***). Therefore, in contrast to the layer-specific alterations in gene expression in the VGlut2 cKO mice, removing IP3R2 impacts astrocytes in all layers and does not strictly follow the developmental trajectory. This suggests a broad requirement for astrocyte calcium signaling in all astrocytes to maintain the correct level of gene expression, and that the signals to do this come from multiple sources and are not restricted to thalamic inputs.

Are there any consequences of diminished astrocyte calcium signaling and altered expression of synapse-regulating genes on synaptic development? As with the VGlut2 cKO, we addressed this by performing immunostaining for presynaptic terminals (VGLUT1 or VGLUT2) and postsynaptic AMPAR subunits (GLUA1 or GLUA2) in Ip3r2 KO mice and WT controls in L1 of VC at P14. We found that the numbers of both cortico-cortical presynaptic terminals marked by VGLUT1, and thal-amocortical terminals marked by VGLUT2, are significantly decreased in Ip3r2 KO mice compared to WT, demonstrating a global deficit in synapse formation in the absence of astrocyte calcium signaling (VGLUT1 from GLUA1: FC 0.86 ± 0.04; ***Figure 5I and J***; VGLUT1 from GLUA2: FC 0.87 ± 0.05; ***Figure 5M and N***. VGLUT2 from GLUA1: FC 0.89 ± 0.05; ***Figure 5—figure supplement 1H and I***; VGLUT2 from GLUA2: FC 0.80 ± 0.05; ***Figure 5—figure supplement 1L and M***). For GLUA1 containing AMPARs, which are regulated by *Gpc4*, we found a significant decrease in the total number of puncta and their colocalization with VGLUT1 and VGLUT2 in the Ip3r2 KO, correlating with the observed decrease in *Gpc4* mRNA (VGLUT1-GLUA1: GLUA1 FC 0.84 ± 0.03; ***Figure 5I and K***; Coloc FC 0.70 ± 0.03; ***Figure 5I and L***; VGLUT2-GLUA1: GLUA1 FC 0.83 ± 0.03; ***Figure 5—figure supplement 1H and J***; Coloc FC 0.74 ± 0.03; ***Figure 5—figure supplement 1H and K***). For GLUA2 containing AMPARs, which are regulated by *Chrdl1*, we found a significant increase in their number, correlating with the observed increase in *Chrdl1* (VGLUT1-GLUA2: GLUA2 FC 1.22 ± 0.07; ***Figure 5M and O***; VGLUT2-GLUA2: GLUA2 FC 1.14 ± 0.03; ***Figure 5—figure supplement 1L and N***). The number of colocalized presynaptic puncta and GLUA2 is, however, unchanged, likely due to the opposing decrease in presynaptic puncta and increase in GLUA2 (VGLUT1-GLUA2 Coloc FC 1.09 ± 0.1; ***Figure 5M and P***; VGLUT2-GLUA2 Coloc FC 1.00 ± 0.08; ***Figure 5—figure supplement 1L and O***).

These results show that GLUA1 and GLUA2 levels are altered in *Ip3r2* KO VC in the direction which follows the change in astrocytic expression of *Gpc4* (which recruits GLUA1) and *Chrdl1* (which recruits GLUA2). This strongly suggests that both astrocytes and neurons play an important role in regulating the expression of synapse-regulating genes, and subsequently AMPAR subunit protein levels, and the final expression levels arise from the complex interaction between these two cell types.

## Unbiased determination of astrocyte transcriptomic diversity and activity-regulated genes in the developing VC

Having found that multiple synapse-regulating genes in astrocytes show layer-specific enrichment, and that these patterns are regulated by neuronal and astrocyte activity, we next asked if these findings are specific to synapse development, or if other astrocyte genes show a similar pattern. To address this using an unbiased approach, we performed single-nucleus RNA sequencing of glial cells isolated from the P14 VC of wild type, VGlut2 cKO, and Ip3r2 KO mice. To isolate the glial cell populations, we immunostained VC nuclei in suspension with an antibody against the neuronal marker NeuN and performed FACS to select the NeuN-negative population (*Figure 6A*). We used the Chromium 10X system to isolate individual glial nuclei and performed RNA sequencing to quantify mRNA levels (*Figure 6A*). This identified 22,781 cells in the VGlut2 condition (cKO and WT) (*Figure 6B*, *Figure 6—figure supplement 1A*) and 21,240 cells in the Ip3r2 condition (KO and WT) (*Figure 6—figure supplement 1B and C*). Initial clustering analysis determined 17 distinct cell populations in both models, with the majority of cells detected clustered within the main glial cell types: astrocytes, microglia, and oligodendrocyte lineage cells (*Figure 6B*, *Figure 6—figure supplement 1A–C*). Just two clusters enriched for neuronal markers are present (*Hernandez et al., 2019*), showing that the NeuN depletion had been successful. This dataset is available in a searchable format online (https://cells.ucsc.edu/?ds=mouse-astro-dev; *Speir et al., 2021*).

## Astrocytes in the wild-type VC form transcriptomically diverse populations

We focused our downstream analysis on astrocytes. A second round of unbiased clustering of the astrocyte population identified four groups (*Figure 6C*, *Figure 6—figure supplement 2A*) in each model and genotype. By comparing the genes enriched in each cluster with datasets in the literature, we determined these to anatomically correspond to upper (L1–2/3), middle (mid; L2/3–5), deep (L5–6) layer, and white matter (WM) astrocytes (*Figure 6C*, *Figure 6—figure supplement 2A*; *Batiuk et al., 2020*; *Bayraktar et al., 2020*; *Lanjakornsiripan et al., 2018*). Similar cell numbers were identified in each group across models and genotypes (number of nuclei in VGlut2 cKO model: upper 1114 WT; 1065 cKO, mid 1289 WT; 1163 cKO, deep 440 WT; 466 cKO, WM 273 WT; 268 cKO; *Figure 6—figure supplement 2C*). We also determined the fractions of astrocytes present in each group and found that this corresponds to the fractions we identified via anatomical cell counts (*Figure 6—figure supplement 2D*), showing that the process of nuclear isolation has captured astrocytes in levels that reflect their in vivo abundance. To validate the layer-enriched genes, we took advantage of publicly available in situ hybridization datasets generated by the Allen Brain Institute (*Allen Brain Institute, 2008*; available from https://developingmouse.brain-map.org/). Cross-referencing cluster-enriched genes identified in our study with P14 in situ hybridization datasets, we found that the gene *Kcnd2* is expressed in mid cortical layers, *Id3* is enriched in deep layers, and *Gfap* is enriched in white matter, matching the snRNAseq (*Figure 6—figure supplement 2B*).

Pairwise comparisons between each of the clusters using the WT astrocytes of each model as the input cells showed that the most robust differences are between the deep and upper layer astrocytes, with over 700 DEGs and up to sixfold log2 FC in expression level (*Figure 6D*, *Figure 6—figure supplement 2E*, *Figure 6—source data 1A and B*). On the other hand, upper and mid astrocytes are the most similar, with about 200 DEGs and twofold log2 FC maximal difference in mRNA level (*Figure 6D*, *Figure 6—figure supplement 2E*, *Figure 6—source data 1A and B*). GO terms analysis of genes enriched in each cluster identified between 300 and 600 terms significantly enriched per cluster, with 140 terms that are common to all four clusters, and between 40 and 120 terms that are unique to each cluster (*Figure 6—figure supplement 2F*, *Figure 6—source data 1C*). Terms with the highest gene ratio in the upper astrocyte cluster include pathways related to signal transduction and membrane biogenesis (*Figure 6—figure supplement 2G*), whereas mid astrocytes are enriched in terms related to GABAergic signaling and PSD95 clustering. Genes belonging to the deep astrocyte cluster are enriched in GO terms related to the regulation of pre- and postsynapse organization (*Figure 6—figure supplement 2F*), while WM astrocyte genes are enriched with pathways related to axonal guidance, maintenance, and signaling (*Figure 6—source data 1C*). Importantly, while there are differences in astrocyte gene expression across layer groups, these are mostly gradients of gene

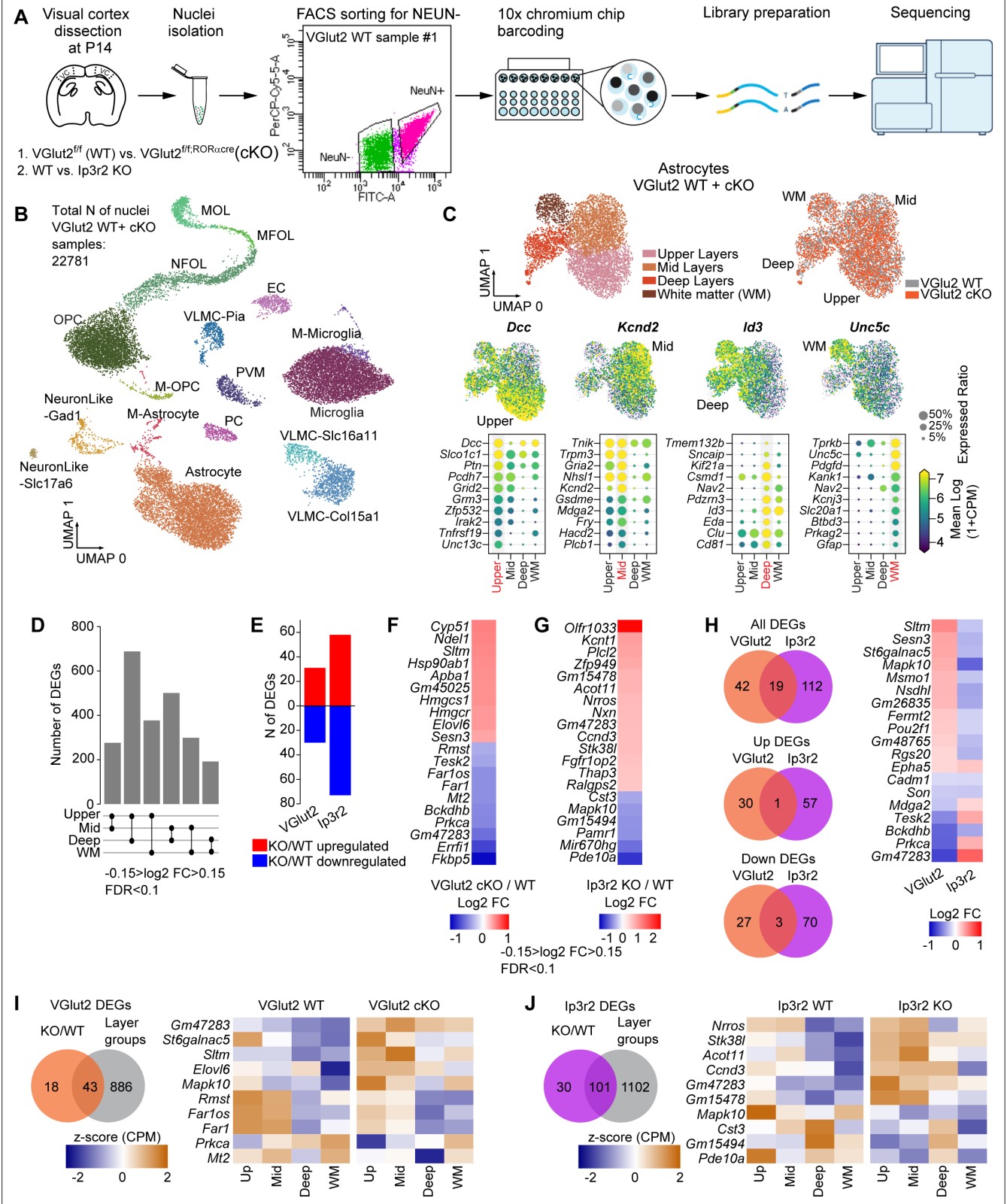

**Figure 6.** Unbiased determination of astrocyte transcriptomic diversity and activity-regulated genes in the developing visual cortex (VC). See also *Figure 6—figure supplements 1–3*, *Figure 6—source data 1*. (**A**) Outline of experiment: VCs collected from VGlut2 cKO, Ip3r2 KO, and their respective WT controls at postnatal day (P)14. Nuclei were isolated from VCs and sorted for NEUN-negative population (glia) using flow cytometry. Sorted nuclei were loaded onto 10X Chromium chip, each nucleus barcoded, followed by library preparation and sequencing. N = 8 samples total. (**B**)

*Figure 6 continued on next page*

*Figure 6 continued*

UMAP clustering of different cell types identified in the NEUN-negative population of the combined samples from VGlut2 WT and cKO mice. 17 clusters were identified including the three main types of glia: astrocytes, oligodendrocytes, and microglia, as well as endothelial cells, and two subtypes of neurons. (M-astrocyte: mitotic astrocyte; M-OPC: mitotic oligodendrocyte precursor cell; OPC: oligodendrocyte precursor cell; MFOL: myelin-forming oligodendrocyte; NFOL: newly formed oligodendrocyte; MOL: mature oligodendrocyte; VLMC: vascular and leptomeningeal cell; EC: endothelial cell; PC: pericyte; PVM: perivascular macrophage). (**C**) Unbiased clustering analysis identified four subpopulations of astrocytes in the P14 VC. Upper panels: left – UMAP plots of astrocyte populations annotated to upper, mid, deep, and white matter types following comparison with published datasets. Right panel – UMAP showing similar clustering obtained for WT and VGlut2 cKO groups. Lower panel shows the expression level of select marker genes that label a particular population as indicated. Each dot represents a single nucleus, color represents expression level in log2 counts per million reads mapped (CPM). Below are dot plots showing a select list of 10 genes that are highly expressed in each population as indicated. Size of the circle is expression ratio (percent cells expressing the gene); color is expression level (log2 CPM). (**D**) Pairwise comparison identified ~200–700 differentially expressed genes (DEGs) between astrocyte populations from WT mice from the VGlut2 cKO model. Larger numbers of DEGs are obtained when comparing upper and deep astrocyte populations. Criteria for DEG selection: log2 fold change (FC) between –0.15 and 0.15; false discovery rate (FDR) < 0.1; see also *Figure 6—source data 1A and B*. (**E–H**) Neuronal and astrocyte activity perturbation results in gene expression changes in astrocytes. (**E**) Number of DEGs identified for each model: VGlut2 cKO: 61 total DEGs; Ip3r2 KO: 131 total DEGs as labeled. Red: upregulated; blue: downregulated. (**F, G**) Heatmap showing top 20 DEGs identified in each model (**F**, VGlut2 cKO; **G**, Ip3r2 KO). Colors represent log2 FC between each condition. Criteria for DEG selection: log2 FC between –0.15 and 0.15; FDR < 0.1. See also *Figure 6—source data 1D*. (**H**) Venn diagrams show number of DEGs common to both models. Heatmap shows FC of the 19 common DEGs. Most common DEGs are inversely regulated in each model (upregulated in VGlut2 cKO and downregulated in Ip3r2 KO). (**I**) Venn diagram showing DEGs common to the VGlut2 cKO vs. WT comparison and genes enriched in astrocyte layer groups. Heatmap of expression level z score of a select list of 10 genes shows dysregulation of layer enrichment in the cKO mice compared to WT. Z-score was calculated for each gene using the combined data for WT and cKO average and standard deviation. (**J**) Same analysis as (**I**), but for the Ip3r2 KO model. See also *Figure 6—source data 1E*.

The online version of this article includes the following figure supplement(s) for figure 6:

**Source data 1.** Unbiased determination of astrocyte layer-enriched genes and global astrocyte gene expression changes following silencing of neuronal or astrocyte activity.

**Figure supplement 1.** Unbiased determination of astrocyte transcriptomic diversity and activity-regulated genes in the developing visual cortex (VC).

**Figure supplement 2.** Unbiased determination of astrocyte transcriptomic diversity and activity-regulated genes in the developing visual cortex (VC).

**Figure supplement 3.** Unbiased determination of astrocyte transcriptomic diversity and activity-regulated genes in the developing visual cortex (VC).

expression, suggesting that the astrocyte layer groups are on a continuum rather than distinct cell types (*Figure 6C*, *Figure 6—figure supplement 2A*; *Bayraktar et al., 2020*; *John Lin et al., 2017*).

Next, we asked how astrocyte marker, function, and synapse-regulating genes highlighted in the bulk RNAseq dataset (*Figure 1G and H*) are expressed across layers (*Figure 6—figure supplement 2I* and J). Overall, we found a positive correlation between levels of gene expression obtained by the two sequencing methods, meaning that genes that were shown to be highly expressed in the bulk dataset (such as *ApoE*) were also highly expressed in the snRNAseq dataset (*Figure 6—figure supplement 2I*). Unsurprisingly, sequencing of bulk RNA samples was more sensitive in detecting the low expressed genes, such as *Gpc4*, *Tgfb1*, and *Thbs1*, which were close to the detection threshold in the snRNAseq dataset, precluding statistical analysis. Nevertheless, plotting the total counts per million mapped reads (CPM) levels for *Gpc4*, *Gpc6*, and *Chrdl1* revealed that they matched the spatial analysis performed at the same age (*Figure 3*). *Gpc4* expression is lowest in the upper layer cluster, *Chrdl1* expression is highest in the upper layer cluster, while *Gpc6* levels are similar across all four populations (*Figure 6—figure supplement 2J*). For other astrocyte marker, function, and synapse-regulating genes detected in the snRNAseq dataset, most exhibited similar levels of expression in all layer groups, with some notable exceptions. For example, *Gfap* and *Aqp4* expression is higher in deep and WM astrocytes than in upper and mid groups, while the expression of connexin 43 (*Gja1*) is highest in deep layer astrocytes compared to all other groups (*Figure 6—figure supplement 2I*). Taken together, these results show that wild-type astrocytes are transcriptomically diverse, but not distinct, in the developing VC, in accordance with previous studies in which astrocyte diversity was assessed at a similar developmental stage (*Bayraktar et al., 2020*).

## Neuronal and astrocyte activity induces global transcriptomic changes in astrocytes

Given that we found that astrocyte synapse-regulating genes are regulated by both neuronal and astrocyte activity, we next asked what other astrocyte genes are affected by these activity manipulations.

To increase the power of our analysis, we combined the four astrocyte subpopulations into one group for each genotype and used this combined group to identify DEGs between the WT and KO. We found 61 DEGs for the VGlut2 cKO model and 131 DEGs for the Ip3r2 KO model (*Figure 6E–H*, *Figure 6—source data 1D*). Performing the same analysis on two other abundant glial populations, OPCs and microglia, showed 28 DEGs for OPCs and 24 DEGs for microglia in the VGlut2 cKO model, and 38 DEGs for OPCs and 29 DEGs for microglia in the Ip3r2 KO model (not shown), 20–50% of the astrocyte DEG level. This suggests that astrocytes are more sensitive to neuronal activity changes, as well as more profoundly affected by silencing their calcium activity. GO analysis of astrocyte DEGs in both models revealed a broad range of BPs, which go beyond terms that may be associated with synapse regulation, such as 'response to stimulus,' 'cell communication,' or 'retrograde axonal transport.' For example, DEGs in the VGlut2 cKO model are also enriched for BP terms related to nuclear envelope disassembly, and nitric oxide metabolism in the upregulated genes, and copper and zinc ion processing in the downregulated genes (*Figure 6—figure supplement 3A*, *Figure 6—source data 1G*). In the Ip3r2 KO model, upregulated genes are enriched for organic acid biosynthesis, while downregulated genes are enriched in histone modification-related BPs (*Figure 6—figure supplement 3B*, *Figure 6—source data 1G*).

Next, as our smFISH analysis of *Gpc4*, *Gpc6*, and *Chrdl1* showed that they are regulated in opposite ways by neuronal and astrocyte activity (*Figures 4 and 5*), we asked whether the DEGs identified by snRNAseq show any overlap between the two models. We found a total of 19 DEGs common to both the VGlut2 cKO and Ip3r2 KO models (*Figure 6H*, *Figure 6—source data 1D*). Only one gene was commonly upregulated, and only three genes commonly downregulated (21% of total common DEGs), while the remaining 15 genes showed opposing changes, similar to *Gpc4* and *Chrdl1* (79% of common DEGs; *Figure 6H* heatmap). The same effect of opposing changes is observed when the overlap between the enriched GO terms was compared (*Figure 6—figure supplement 3C*). For example, the term 'intracellular signal transduction' (GO:0035556) is enriched in the downregulated gene list in the VGlut2 cKO model, whereas in the Ip3r2 KO model it is enriched in the upregulated gene list (*Figure 6—figure supplement 3C*, *Figure 6—source data 1G*).

Since we found that neuronal but not astrocyte activity is important for regulating layer-specific expression of *Gpc4*, *Gpc6*, and *Chrdl1* (*Figure 2*, *Figure 4* and *Figure 5*), we next asked if this is true for other DEGs identified in the snRNAseq dataset in each model (*Figure 6I and J*). To address this, we compared the list of neuronal or astrocyte activity-regulated genes with the list of layer-enriched genes from the WT (*Figure 6I and J*, *Figure 6—source data 1E*) and identified 43 common genes in the VGlut2 cKO model (70% of all cKO/WT DEGs; *Figure 6I*) and 101 common DEGs in the Ip3r2 model (77% of all KO/WT DEGs; *Figure 6J*). In contrast to our expectation of finding layer-dependent expression changes only in the VGlut2 cKO model, we found that in each model a subset of genes show a dysregulated layer expression when activity is altered. For example, expression of *Mapk10* is lowest in WM astrocytes in the WT; however, in astrocytes from the VGlut2 cKO *Mapk10* is upregulated in WM astrocytes compared to deep (*Figure 6I*). Similarly, the expression of *Pde10a* in the Ip3r2 KO model WT group is higher in the upper astrocyte group than other groups, while in the KO the level in that group is now low (*Figure 6J*). WT expression of the gene *Gm47283* is higher in the upper than deep layer group, while in the VGlut2 cKO, *Gm47283* is upregulated in all groups, and the relative expression between layers is maintained (*Figure 6I*). A similar pattern is observed in the Ip3r2 KO model for the gene *Stk38l* (*Figure 6J*). Thus, perturbation of neuronal or astrocyte activity influences the layer enrichment of some but not all genes, without causing gross rearrangement of overall astrocyte spatial identity.

Further, while the expression of *Gpc4*, *Gpc6*, and *Chrdl1* shows opposite levels in VGlut2 cKO mice to those seen during normal development (*Figure 4*), astrocyte activity does not regulate these genes in the same way (*Figure 5*). To test if this is a general phenomenon beyond synapse-regulating genes, we took advantage of the bulk RNAseq dataset (*Figure 1*, *Figure 1—source data 1A*) and calculated the FC in gene expression of WT astrocytes between the time points P7 and P14 (FC P14/P7; *Figure 6—figure supplement 3D and E*, *Figure 6—source data 1F*). We compared the DEG lists of VGlut2 cKO and Ip3r2 KO models against the genes that are significantly up- or downregulated at P14 compared to P7 in the bulk data in search for common genes (*Figure 6—figure supplement 3D and E*). This identified 30 DEGs from the VGlut2 cKO dataset (52% of all cKO/WT DEGs, *Figure 6—figure supplement 3D*) and 57 genes from the Ip3r2 KO dataset (44% of all KO/WT DEGs, *Figure 6—figure*

*supplement 3E*). Comparing the direction of expression changes between normal development and the VGlut2 cKO DEGs, we observed that 70% of common genes were regulated in the opposite direction (*Figure 6—figure supplement 3D*, middle panel), similar to *Gpc4*, *Gpc6*, and *Chrdl1* (*Figure 4*), suggesting strong dependence of astrocyte developmental maturation on neuronal cues. On the other hand, Ip3r2 KO data showed 50% of genes displaying the same directionality as during normal development, and the other 50% showing the opposite regulation (*Figure 6—figure supplement 3E*). Taken together, these results show that during development astrocyte-neuron communication enacts global gene expression changes in astrocytes, going beyond synapse-regulating genes.

## Discussion

In this study, we demonstrate how astrocytes and synapses develop together in the postnatal mouse VC, signaling to one another to ensure correct development (*Figure 7*). In particular, we show that:

- expression of select synapse-regulating genes (*Gpc4*, *Gpc6,* and *Chrdl1*) is differentially regulated during development at both temporal and spatial levels
- astrocyte transcriptome changes during development are correlated to expression changes in some synaptic proteins
- expression of astrocyte synapse-regulating genes is affected by changes in thalamic neuronal activity and astrocyte calcium activity
- astrocytes form heterogeneous populations in the developing cortex, based on their spatial location
- neuronal and astrocyte activity regulates multiple non-overlapping genetic programs in astrocytes, suggesting effects beyond synapse regulation.

### Astrocyte number and transcriptome alterations across development coincide with stages of synapse development

In the mouse cortex, astrocytes begin to be generated right before birth and populate the cortex throughout the first month of life (*Farhy-Tselnicker and Allen, 2018*; *Ge et al., 2012*). During this time, many changes are occurring in astrocytes, as well as in the synapses between neighboring neurons. We observed that the most significant change in astrocytes at the transcriptome level occurred between the first and second postnatal weeks (*Figure 1*). Similarly, an analysis of the synaptic proteome during development showed the largest difference between P9 and P15 (*Gonzalez-Lozano et al., 2016*), suggesting similar or overlapping regulatory mechanisms in both astrocytes and neurons. In addition to the transcriptomic changes, astrocyte numbers are also strongly regulated during development. Indeed, genes upregulated at P14 are uniquely enriched in GO terms related to cell proliferation and migration (*Figure 1—source data 1B*). Interestingly, the density of astrocytes remains fairly constant throughout development, suggesting that their expansion rate is correlated with the overall expansion of the brain tissue. The mechanisms that regulate these migration patterns are still unknown and seem to be largely unaffected by neuronal or astrocyte activity, as evident from the similar numbers of astrocytes within each cortical layer in both neuronal and astrocyte activity manipulation models tested here (*Figure 4—figure supplement 2*, *Figure 5—figure supplement 1*). Future studies will determine the factors or sets of factors that regulate the number and location of astrocytes within defined domains.

### Astrocytes form diverse populations in the developing mouse VC

The diversity of neurons based on location, morphology, connectivity, and activity patterns has been extensively studied for decades, with multiple subtypes of excitatory and inhibitory neurons identified (*Kepecs and Fishell, 2014*; *Migliore and Shepherd, 2005*; *Zeisel et al., 2015*). For a long time, cortical protoplasmic astrocytes were viewed as a homogeneous population. However, recent studies looking in-depth at astrocyte heterogeneity using both bulk and single-cell sequencing approaches have shown that within the cortex astrocytes form a heterogeneous population (*Batiuk et al., 2020*; *Bayraktar et al., 2020*; *Lanjakornsiripan et al., 2018*). Unlike neurons, astrocytes do not fall into the six-layer categories, but rather exist on a gradient of transcriptomically separable yet overlapping groups. Indeed, our snRNAseq data shows that the biggest differences are between astrocytes of the upper and deep layer groups, while upper and mid-layer groups are the most similar (*Figure 6C*

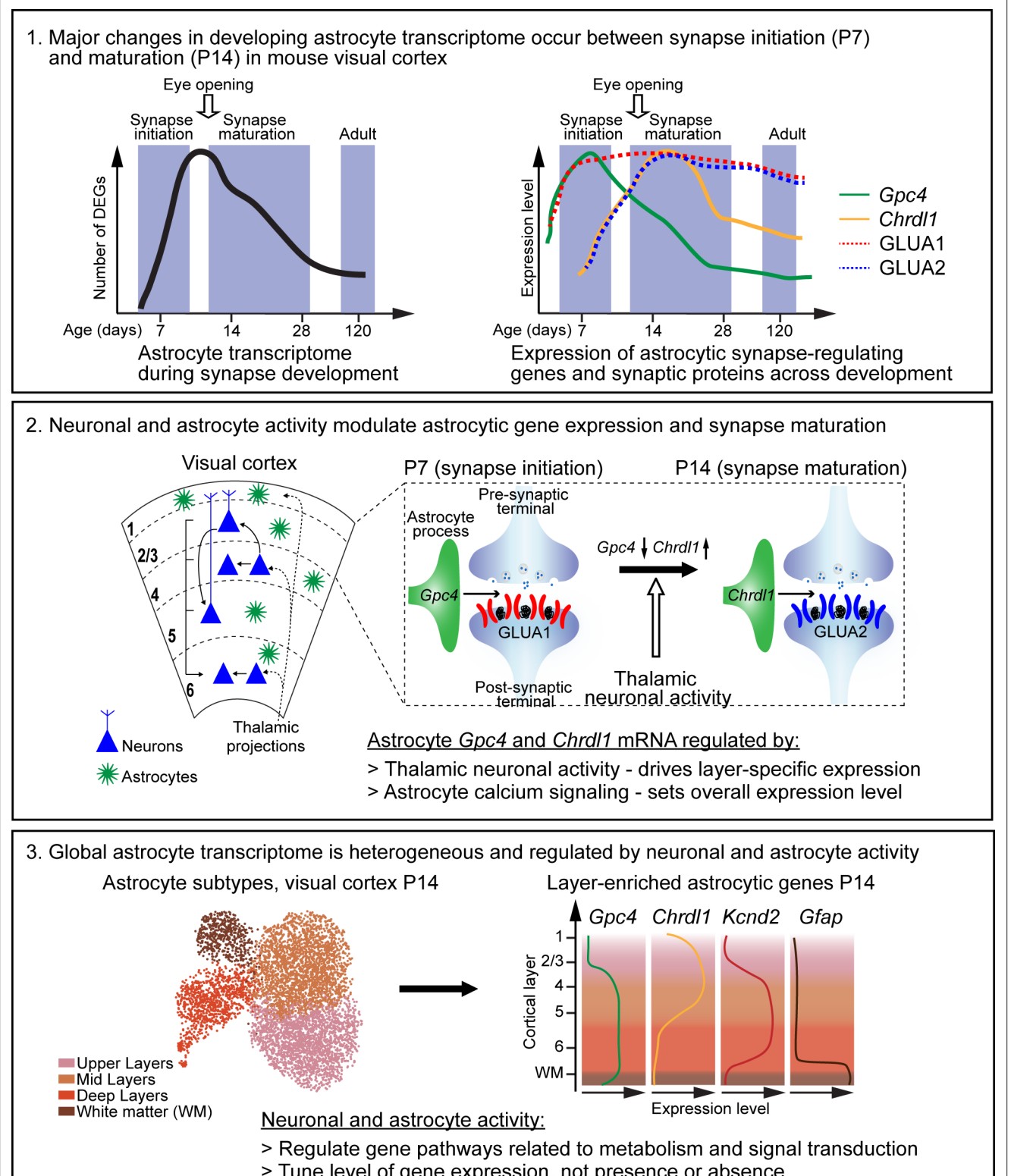

**Figure 7.** Summary of key findings and working model. Box 1. Astrocytes and synapses develop across a similar time line in the mouse visual cortex (VC). Left diagram: majority of astrocyte transcriptomic changes (represented as number of differentially expressed genes [DEGs]) occur between postnatal day (P)7 (synapse initiation) and P14 (synapse maturation). Right diagram: select astrocytic synapse-regulating genes (*Gpc4* and *Chrdl1*) and synaptic proteins (GLUA1, GLUA2) are differentially expressed between P7 and P14. Box 2. The spatio-temporal expression of astrocyte synapse-

*Figure 7 continued on next page*

*Figure 7 continued*

regulating genes is driven by thalamic neuronal activity and astrocyte calcium activity. Left: diagram of VC depicting neuronal (blue) laminar arrangement and connectivity (arrows). Astrocytes (green) are present in all cortical layers. Right: expression of astrocytic *Gpc4* and *Chrdl1* mRNA, and their regulated synaptic GLUAs during development. *Gpc4* expression is decreased at P14, *Chrdl1* expression is increased (correlating with increase in GLUA2 subunits, which are regulated by *Chrdl1*). These changes are regulated by thalamic neuronal activity in a time- and layer-specific manner. Additionally, overall expression of astrocytic genes is regulated by astrocyte calcium signaling. Box 3. Single-cell transcriptomic profiling of VC astrocytes at P14 reveals a heterogeneity of gene expression, with a global transcriptomic dependence on neuronal and astrocyte activity. Left: UMAP plot shows four different astrocyte clusters identified in the wild-type VC, which correspond to spatial organization in the cortex. Right: diagram showing layer-specific expression of select astrocytic genes as labeled.

*and D*, *Figure 6—figure supplement 2A and E*). Nevertheless, we have identified several astrocyte population marker genes (such as *Dcc* or *Kcnd2*; *Figure 6*, *Figure 6—figure supplement 2B*, *Figure 6—source data 1A and B*), which are significantly enriched in one group over others. Before performing functional studies based on these genes, further characterization is required, for example, cross-referencing these genes with our bulk RNA sequencing dataset to identify astrocyte-enriched genes and performing immunohistochemistry (IHC) to determine if protein expression is also heterogeneous. These validated genes could then be used to target specific populations of astrocytes, similar to the methods employed for neurons, in order to manipulate astrocytes that interact with specific synapse types or circuits. Importantly, while blocking thalamocortical activity did alter the expression of multiple genes in astrocytes, it did not alter the layer patterning of the cells, showing that this is not a major factor in driving layer-enriched gene expression. Indeed, altering the identity of local cortical neurons by using *Dab1* KO mice, in which cortical layer neurons are reversed, does alter astrocyte layer identity, suggesting a role for local cues (*Lanjakornsiripan et al., 2018*). Our findings further suggest that neuronal activity acts to fine-tune the level of astrocyte genes that are important for neuronal function, rather than determining their presence or absence. Functional studies are further needed to identify the precise neuronal activity patterns that govern astrocyte-neuron reciprocal communication.

Moreover, some of the synapse-regulating genes we profiled display layer-specific expression changes across development (*Figure 2*). We found a correlation between *Chrdl1* upregulation in the upper layers with that of GLUA2, consistent with our previous findings regarding *Chrdl1* regulation of GLUA2 levels (*Blanco-Suarez et al., 2018*). A more complex picture emerges for *Gpc4*, *Gpc6*, and GLUA1, the AMPAR subunit regulated by these factors (*Allen et al., 2012*). GLUA1 protein levels steadily increase across development, peaking in most layers at P7, and do not show downregulation at P14 in L1 (as was observed for *Gpc4*), or upregulation in deeper layers (as was shown for *Gpc6*). Still, changes in *Gpc4* expression are contributing to the levels of GLUA1, as GLUA1 is affected in correlation with changes in *Gpc4* expression in the neuronal and astrocyte activity-deficit models (*Figures 4 and 5*), and GLUA1 levels are reduced in the VC of *Gpc4* KO mice (*Farhy-Tselnicker et al., 2017*). One possibility is that *Gpc4* and *Gpc6* may regulate GLUA1 levels at specific synapses, such as glutamatergic terminals onto interneurons in L1, or deep layer cortical neurons, making it hard to distinguish their specific effect when analyzing synapses as a group. Alternatively, they may be required to induce initial recruitment of GLUA1 to synaptic sites, but not for its maintenance, so GLUA1 levels remain stable when *Gpc4* is downregulated with development.

## Neuronal and astrocyte activity modulate transcriptomic changes in astrocytes

Ever since the astrocyte-derived factors that promote synapse formation were identified, an outstanding question in the field has been, how are they regulated (*Baldwin and Eroglu, 2017*; *Farhy-Tselnicker and Allen, 2018*)? Is it astrocyte-intrinsic, or is it driven by changes in neuronal activity that occur as synapses develop? Our in vitro work, together with previously published studies, has provided evidence that neuronal activity can influence astrocyte gene expression and function at the synapse (*Figure 4—figure supplement 1*, *Benediktsson et al., 2012*; *Bernardinelli et al., 2014b*; *Durkee and Araque, 2019*; *Hasel et al., 2017*). However, how this occurs in the developing brain in vivo has not been systematically addressed. Here, we approached this question by perturbing the activity of thalamocortical projections (through VGlut2 knockout), with a goal of manipulating signals that have the potential to regulate astrocytic developmental gene

expression under physiological conditions. Indeed, this perturbation resulted in attenuation of the developmental expression changes in astrocyte genes as well as AMPAR subunits at P14 but not at P7, suggesting a disruption in circuit maturation. A similar outcome on neuronal and astrocyte maturation was observed in studies employing visual deprivation methods (*Albanese et al., 1983*; *Desai et al., 2002*; *Freire, 1978*; *Funahashi et al., 2013*; *Ishikawa et al., 2014*; *Ko et al., 2014*; *Müller, 1990*; *Stogsdill et al., 2017*). Future strategies including manipulation of neuronal and astrocyte function using opto- or chemogenetic approaches will further elucidate the role of astrocyte-neuron interaction in circuit development and maturation.

An additional important regulation of astrocyte gene expression (including *Gpc4* and *Chrdl1*) revealed here is by IP3R2-mediated astrocyte calcium activity, a central mechanism for astrocytic somal calcium increases, and as such, a likely candidate to modulate gene expression changes (*Petravicz et al., 2014*; *Srinivasan et al., 2015*; *Yu et al., 2020*). Interestingly, blunting store-mediated astrocytic calcium release (through Ip3r2 knockout) resulted in an opposite regulation of gene expression to the ones observed in the VGlut2 cKO mice and did not correspond to layer-specific developmental changes, suggesting a more global role of astrocyte calcium activity in the regulation of gene expression. This is likely due to the overall diminished signaling that occurs in this knockout model in response to multiple neurotransmitters that rely on calcium increases, and in future, systematic deletion of individual receptors will be necessary to tease out the impact of each neurotransmitter on astrocyte gene expression and synapse regulation. Notably, while we observed significant effects on *Gpc4* and *Chrdl1* mRNA levels, *Gpc6* expression was unaltered in Ip3r2 KO VC at P14, suggesting distinct regulation of expression of the two glypican family members, and synapse-modulating genes in general. The mechanisms underlying these effects are unknown and likely involve activation of $Ca^{2+}$-dependent cellular cascades and transcription factors. Indeed, several such genes were identified in our snRNAseq dataset including *Mapk10* and *Pde4b*, and will form the focus of future studies. Previous work in the adult striatum found that increasing astrocytic calcium through activation of exogenous Gi-GPCRs using DREADDs induces transient synapse formation by regulating expression of thrombospondin 1, with no effect on *Gpc4* or *Chrdl1* (*Nagai et al., 2019*), suggesting specific dependence of age, region, and environmental cues on types of genes regulated by calcium activity. Furthermore, in addition to altered gene expression in astrocytes, we observed diminished levels of presynaptic markers VGLUT1 and VGLUT2 in Ip3r2 KO mice, suggesting that in addition to having intrinsic roles in regulating astrocytic gene expression IP3R2-mediated activity has a non-cell-autonomous effect on nearby neurons. These data may suggest that behavioral deficits observed in Ip3r2 KO mice could stem from the developmental alterations in synaptogenesis caused by lack of astrocyte calcium responses. Importantly, the activity-mediated regulation of astrocytic gene expression observed here appears to extend beyond synapse development as DEGs identified in the snRNAseq analysis of VGlut2 cKO and Ip3r2 KO mice span multiple cellular processes, including signal transduction, metabolism, and gene regulation (*Figure 6—figure supplement 3A–C*, *Figure 6—source data 1G*). Many of these genes may potentially be involved in regulating synapses, and further studies are required to characterize the most promising candidate genes identified here (such as *Sltm*, *Gm47283*). This will help to determine the precise role of thalamocortical and astrocytic calcium-dependent transcriptomic changes in regulating the different aspects of both astrocyte and synapse function.

In all, this study demonstrates that the correct formation of synapses and hence neuronal circuit connectivity depends on precise communication between neurons and astrocytes, where disruption in one cell type leads to disruption in the other and an overall dysregulation of synapse formation. It further shows that astrocyte regulation of synapses is intimately linked to environmental changes. Thus, an image of astrocyte identity emerges as highly plastic and dynamic cells, actively perceiving and responding to their environment. Future studies employing functional approaches such as electrophysiology, optogenetic manipulations, and behavior are needed to determine the precise nature of astrocyte plasticity and to further distinguish intrinsic and extrinsic influences on these cells, giving further insight into their function in both health and disease.

## Materials and methods

**Key resources table**

| Reagent type (species) or resource | Designation | Source or reference | Identifiers | Additional information |
|---|---|---|---|---|
| Antibody | Anti-HA tag (rat monoclonal) | Roche | CAT# 11867423001; RRID:AB_390918 | IF (1:1000) |
| Antibody | Anti-HA tag (rabbit monoclonal) | CST | CAT# 3724; RRID:AB_1549585 | RiboTag pulldown (1:200) IF (1:500) |
| Antibody | Anti-NeuN (mouse monoclonal) | Millipore | CAT# MAB377; RRID:AB_2298772 | IF (1:100) |
| Antibody | Anti-S100β (rabbit polyclonal) | Abcam | CAT# ab52642; RRID:AB_882426 | IF (1:100) |
| Antibody | Anti-Ng2 (rabbit polyclonal) | Millipore | CAT# Ab5320; RRID:AB_11213678 | IF (1:200) |
| Antibody | Anti-MOG (rabbit polyclonal) | Proteintech | CAT# 12690-1-ap; RRID:AB_2145527 | IF (1:200) |
| Antibody | Anti-Iba1 (rabbit polyclonal) | Wako | CAT# 016-20001; RRID:AB_839506 | IF (1:250) |
| Antibody | Anti-Sox9 (rabbit monoclonal) | Abcam | CAT# ab185966; RRID:AB_2728660 | IF (1:2000) |
| Antibody | Anti-Aldh1l1 (rabbit polyclonal) | Abcam | CAT# ab-87117; RRID:AB_10712968 | IF (1:500) |
| Antibody | Anti-glypican 4 (rabbit polyclonal) | Proteintech | CAT# 13048-1-AP; RRID:AB_10640157 | WB (1:500) |
| Antibody | Anti-IP3R2 (rabbit polyclonal) | Ju Chen lab UCSD | N/A | WB (1:1000) |
| Antibody | Anti-β-tubulin (mouse monoclonal) | Thermo | CAT# MA5-16308; RRID:AB_2537819 | WB (1:5000) |
| Antibody | Anti-Bassoon (mouse monoclonal) | Enzo | CAT# VAM-PS003; RRID:AB_2066982 | IF (1:500) |
| Antibody | Anti-VGLUT1 (guinea pig polyclonal) | Millipore | CAT# AB5905; RRID:AB_2301751 | IF (1:2000) |
| Antibody | Anti-VGLUT2 (guinea pig polyclonal) | Millipore | CAT# AB2251; RRID:AB_2665454 | IF (1:3000–5000) |
| Antibody | Anti-GLUA1 (rabbit polyclonal) | Millipore | CAT# AB1504; RRID:AB_2113602 | IF (1:400) |
| Antibody | Anti-GLUA2 (rabbit polyclonal) | Millipore | CAT# AB1768-I; RRID:AB_2313802 | IF (1:400) |
| Antibody | Anti-Neun-Alexa-488 (mouse monoclonal) | Millipore | CAT# MAB377X; RRID:AB_2149209 | FACS (1:1000) |
| Antibody | Anti-GFP (chicken polyclonal) | Millipore | CAT# 06-896; RRID:AB_11214044 | IF (1:500) |
| Antibody | Anti-rat Alexa-488 (goat polyclonal) | Molecular Probes | CAT# A11006; RRID:AB_141373 | IF (1:500) |
| Antibody | Anti-rat Alexa-594 (goat polyclonal) | Molecular probes | CAT# A11007; RRID:AB_141374 | IF (1:500) |
| Antibody | Anti-mouse Alexa-488 (goat polyclonal) | Molecular Probes | CAT# A11029; RRID:AB_138404 | IF (1:500) |
| Antibody | Anti-mouse Alexa-594 (goat polyclonal) | Molecular Probes | CAT# A11032; RRID:AB_141672 | IF (1:500) |
| Antibody | Anti-mouse Alexa-680 (goat polyclonal) | Molecular Probes | CAT# A21057; RRID:AB_141436 | WB (1:10,000) |
| Antibody | Anti-rabbit Alexa-488 (goat polyclonal) | Molecular Probes | CAT# A11034; RRID:AB_2576217 | IF (1:500) |

| Reagent type (species) or resource | Designation | Source or reference | Identifiers | Additional information |
|---|---|---|---|---|
| Antibody | Anti-rabbit Alexa-594 (goat polyclonal) | Molecular Probes | CAT# A11037; RRID:AB_2534095 | IF (1:500) |
| Antibody | Anti-rabbit Alexa-647 (goat polyclonal) | Molecular Probes | CAT# A21245; RRID:AB_2535813 | IF (1:500) |
| Antibody | Anti-rabbit Alexa-680 (goat polyclonal) | Molecular Probes | CAT# A21109; RRID:AB_2535758 | WB (1:10,000) |
| Antibody | Anti-guinea pig Alexa-488 (goat polyclonal) | Molecular Probes | CAT# A11073; RRID:AB_142018 | IF (1:500) |
| Antibody | Anti-guinea pig Alexa-594 (goat polyclonal) | Molecular Probes | CAT# A11076; RRID:AB_141930 | IF (1:500) |
| Antibody | Anti-guinea pig Alexa-647 (goat polyclonal) | Molecular Probes | CAT# A21450; RRID:AB_141882 | IF (1:500) |
| Antibody | Anti-chicken Alexa-488 (goat polyclonal) | Molecular Probes | CAT# A11039; RRID:AB_142924 | IF (1:500) |
| Chemical compound, drug | Papain | Worthington | CAT# PAP2 3176 | |
| Chemical compound, drug | Trypsin inhibitor | Worthington | CAT# LS003086 | |
| Chemical compound, drug | Isolectin | Vector | CAT# L-1100 | |
| Chemical compound, drug | Trypsin | Sigma | CAT# T9935 | |
| Chemical compound, drug | Poly-D-lysine | Sigma | CAT# P6407 | |
| Chemical compound, drug | Laminin | Cultrex Trevigen | CAT# 3400-010-01 | |
| Chemical compound, drug | N-acetyl-L-cysteine | Sigma | CAT# A8199 | |
| Chemical compound, drug | Insulin | Sigma | CAT# I1882 | |
| Chemical compound, drug | Triiodo-thyronine | Sigma | CAT# T6397 | |
| Chemical compound, drug | Transferrin | Sigma | CAT# T1147 | |
| Chemical compound, drug | BSA | Sigma | CAT# A4161 | |
| Chemical compound, drug | Progesterone | Sigma | CAT# P6149 | |
| Chemical compound, drug | Putrescine | Sigma | CAT# P5780 | |
| Chemical compound, drug | Sodium selenite | Sigma | CAT# S9133 | |
| Chemical compound, drug | Forskolin | Sigma | CAT# F6886 | |
| Chemical compound, drug | FUDR | Sigma | CAT# F0503 | |
| Chemical compound, drug | AraC | Sigma | CAT# C1768 | |
| Chemical compound, drug | Hydrocortisone | Sigma | CAT# H0888 | |
| Chemical compound, drug | B27/NS21 | *Winzeler and Wang, 2013* | N/A | |
| Peptide, recombinant protein | BDNF | PeproTech | CAT# 450-02 | |
| Peptide, recombinant protein | CNTF | PeproTech | CAT# 450-13 | |
| Chemical compound, drug | Complete Protease Inhibitor Cocktail | Sigma | CAT# 04693132001 | |
| Chemical compound | L-lysine | Sigma | CAT# L5501 | |
| Chemical compound, drug | SlowFade Gold with DAPI mounting media | Thermo Fisher | CAT# S36939 | |
| Chemical compound, drug | DAPI | Millipore | CAT# 5.08741.0001 | |
| Chemical compound, drug | Ketamine | Victor Medical Company | N/A | |

| Reagent type (species) or resource | Designation | Source or reference | Identifiers | Additional information |
|---|---|---|---|---|
| Chemical compound, drug | Xylazine | Anased | N/A | |
| Peptide, recombinant protein | RNAsin | Promega | CAT# N2115 | |
| Chemical compound, drug | Heparin | Sigma | CAT# H3393 | |
| Chemical compound, drug | Protease Inhibitor Cocktail | Sigma | CAT# P8340 | |
| Peptide, recombinant protein | RNaseOUT Recombinant Ribonuclease Inhibitor | Thermo Fisher | CAT# 10777019 | |
| Peptide, recombinant protein | SUPERase• In RNase Inhibitor | Thermo Fisher | CAT# AM2694 | |
| Chemical compound, drug | OptiPrep | Sigma | CAT# D1556 | |
| Chemical compound, drug | Hoechst 33342 | Thermo Fisher | CAT# 62249 | |
| Other | Magnetic IgG beads | Thermo Fisher | CAT# 88847 | |
| Chemical compound, drug | Cycloheximide | Sigma | CAT# C7698 | |
| Chemical compound, drug | UltraPure BSA | Thermo Fisher | CAT# AM2618 | |
| Commercial assay or kit | Bradford assay | Bio-Rad | CAT# 5000203 | |
| Commercial assay or kit | RNeasy mini kit | Qiagen | CAT# 74104 | |
| Commercial assay or kit | RNAeasy plus micro kit | Qiagen | CAT# 74034 | |
| Commercial assay or kit | RNAscope 2.5 HD—multiplex fluorescent Manual Assay | ACDbio | CAT# 320850 | |
| Commercial assay or kit | Qbit | Thermo Fisher | CAT# Q33238 | |
| Commercial assay or kit | TapeStation | Agilent | CAT# G2991AA | |
| Commercial assay or kit | TruSeq Stranded mRNA Library Preparation Kit | Illumina | CAT# RS-122-2101 | |
| Commercial assay or kit | 10X Chromium 3' kit V3 | 10X Genomics | CAT# PN-1000073 | |
| Cell line (mouse) | T11D7e2 Hybridoma | ATCC | CAT# TIB-103; RRID:CVCL_F769 | |
| Genetic reagent (mouse) | Tg(Aldh1l1-EGFP)OFC789Gsat/Mmucd | UC Davis | 011015-UCD; RRID:MMRRC_011015-UCD | |
| Genetic reagent (mouse) | B6N.129-Rpl22tm1.1Psam/J | Jackson Labs | Jax # 011029; RRID:IMSR_JAX:011029 | |
| Genetic reagent (mouse) | B6.Cg-Tg(Gfap-cre)73.12Mvs/J | Jackson Labs | Jax# 012886; RRID:IMSR_JAX:012886 | |
| Genetic reagent (mouse) | B6.C-Gt(ROSA)26Sortm14(CAG-tdTomato) | Jackson Labs | Jax # 007914; RRID: IMSR_JAX:007914 | |
| Genetic reagent (mouse) | Slc17a6tm1Lowl/J | Jackson Labs | Jax # 12898; RRID:IMSR_JAX:012898 | |
| Genetic reagent (mouse) | Rora$^{tm1(cre)Ddmo}$ | O'Leary lab (Salk Institute) *Chou et al., 2013* | | MGI:5000017 |
| Genetic reagent (mouse) | Itpr2$^{tm1.1Chen}$ | Chen lab (UCSD) *Li et al., 2005* | | MGI:3640970 |
| Sequence-based reagent | RNAscope probe: 3-plex negative control | ACDbio | CAT# 320871 | |
| Sequence-based reagent | RNAscope probe: *Gpc4* | ACDbio | CAT# 442821 | |
| Sequence-based reagent | RNAscope probe: *Gpc5* | ACDbio | CAT# 442831 | |
| Sequence-based reagent | RNAscope probe: *Gpc6*-01 | ACDbio | CAT# 453301 | |

| Reagent type (species) or resource | Designation | Source or reference | Identifiers | Additional information |
|---|---|---|---|---|
| Sequence-based reagent | RNAscope probe: *Chrdl1* | ACDbio | CAT# 442811 | |
| Sequence-based reagent | RNAscope probe: *Thbs1* | ACDbio | CAT# 457891 | |
| Sequence-based reagent | RNAscope probe: *Thbs2* | ACDbio | CAT# 492681 | |
| Sequence-based reagent | RNAscope probe: *Thbs4* | ACDbio | CAT# 526821 | |
| Sequence-based reagent | RNAscope probe: Slc1a3 (Glast) channel 2 | ACDbio | CAT# 430781-C2 | |
| Sequence-based reagent | RNAscope probe: Tubb3 channel 3 | ACDbio | CAT# 423398-C3 | |
| Software, algorithm | Imaris | Bitplane | RRID:SCR_007370 | |
| Software, algorithm | ImageJ (Fiji) | NIH | RRID:SCR_003070 | |
| Software, algorithm | Zen | Zeiss | RRID:SCR_013672 | |
| Software, algorithm | AxioVision | Zeiss | RRID:SCR_002677 | |
| Software, algorithm | Odyssey Image Studio | LI-COR | RRID:SCR_014211 | |
| Software, algorithm | InteractiVenn | *Heberle et al., 2015* | N/A | |
| Software, algorithm | GraphPad Prism | Prism | RRID:SCR_002798 | |
| Software, algorithm | Python | | RRID:SCR_008394 | |
| Software, algorithm | RStudio | | RRID:SCR_000036 | |
| Other | Cell culture inserts | Thermo Fisher | CAT# 353102 | |
| Other | Vivaspin centrifugal concentrator | Sartorius | CAT# 14558502 | |
| Other | 4–12% bolt gels | Thermo Fisher | CAT# NW04120 | |
| Other | PVDF membranes, Immobilon-FL | Millipore | CAT# IPFL00005 | |
| Other | Odyssey Infrared Imager | LI-COR | N/A | |
| Other | Fluorescence microscope with apotome | Zeiss | Axio Imager.Z2 | |
| Other | Cryostat | Hacker Industries | OTF5000 | |
| Other | Confocal microscope | Zeiss | LSM710 | |
| Other | Confocal microscope | Zeiss | LSM880 | |
| Other | FACS Aria Fusion sorter | BD | N/A | |
| Other | HybEZ hybridization system | ACDbio | N/A | |
| Other | Illumina HiSeq 2500 | Illumina | N/A | |
| Other | NovaSeq 6000 | Illumina | N/A | |

## Contact for reagent and resource sharing

Further information and requests for resources and reagents should be directed to and will be fulfilled by the lead contact, Nicola J. Allen (nallen@salk.edu).

## Animals

All animal work was approved by the Salk Institute Institutional Animal Care and Use Committee.

### Rats

Sprague–Dawley rats (Charles Rivers) were maintained in the Salk Institute animal facility under a 12 hr light:dark cycle with ad libitum access to food and water. Rat pups (both male and female) were used at P1–2 for preparation of primary cortical astrocyte cultures, and at P5–P7 for preparation of purified immunopanned retinal ganglion cell (RGC) neuronal cultures.

## Mice

Mice were maintained in the Salk Institute animal facility under a 12 hr light:dark cycle with ad libitum access to food and water. Both male and female mice were used for experiments.

The following mouse lines were used:

1. Wild-type (WT; C57Bl6/J) were purchased from Jackson Labs and bred in-house (Jax #000664). Mice were used for breeding and backcrossing, and as non-littermate controls.

2. RiboTag floxed (B6N.129-Rpl22[tm1.1Psam/J]) were obtained from Jackson Labs (Jax #011029). Mice were maintained as homozygous for floxed Rpl22 on C57Bl6/J background and crossed to mice expressing cre recombinase for experiments.

Gfap-cre (B6.Cg-Tg (Gfap-cre)[73.12Mvs/J]) mice were obtained from Jackson Labs (Jax #012886) and bred in-house to generate cre+ females.

To generate Astrocyte-RiboTag mice, homozygous flox-Rpl22-HA males were crossed to Gfap-cre hemizygous females. Male mice hemizygous for cre and heterozygous for flox-Rpl22-HA (Rpl22-HA+; Gfapcre+) were used for all experiments.

3. Aldh1l1-GFP (Tg(Aldh1l1-EGFP)[OFC789Gsat/Mmucd]) were obtained from MMRRC. They were back-crossed to C57Bl6/J background (Jax #000664) for at least four generations prior to conducting experiments.

4. VGlut2 floxed (Slc17a6[tm1Lowl/J]) were obtained from Jackson Labs (Jax #012898). Mice were maintained as homozygous for floxed VGlut2 on a C57Bl6/J background and crossed to mice expressing cre recombinase for experiments.

RORα-IRES-Cre (Rora[tm1(cre)Ddmo]) were obtained from Dennis O'Leary at the Salk Institute and described in *Chou et al., 2013* and *Farhy-Tselnicker et al., 2017*. Mice were maintained on the C57Bl6/J (Jax #000664) background and crossed to the VGlut2 flox and/or tdTomato reporter (B6.Cg-Gt(ROSA)26Sor[tm14(CAG-tdTomato)Hze]/J (Ai14; Jax #007914)) lines for experiments.

To generate conditional VGlut2 KO mice, Slc17a6[f/f] females were bred to Slc17a6[f/f];RORα[cre+]-positive males. Slc17a6[f/f];RORα[cre+] (cre-positive) littermates were compared with Slc17a6[f/f];RORα[cre-] (cre-negative) in each experiment.

To generate het and homozygous VGlut2 cKO mice expressing tdTomato in the recombined neurons (*Figure 4*, *Figure 4—figure supplement 2*), Slc17a6[f/+];RORα[cre+];Rosa26[tdTomato+] males were crossed to Slc17a6[f/f] females. As control in these experiments, RORα[cre+] mice were crossed to Rosa26[tdTomato+], generating Slc17a6[+/+];RORα[cre+]Rosa26[tdTomato+] mice.

5. Ip3r2 KO (Itpr2[tm1.1Chen]) was obtained from Ju Chen lab at UCSD (*Li et al., 2005*) and maintained on C57BL6/J background, either as KO × KO breeding scheme, or het × het breeding scheme. Both littermate and non-littermate pairs of WT and KO mice were used for experiments. For non-littermate pairs, C57Bl6/J that were bred in-house were used as control.

In all cases, when littermates could not be used as control, mice were matched by age, size, fur color and condition, and eye opening to ensure identical developmental stage.

## Mouse tissue collection

Tissue was collected at the following developmental time points: P1, P4, P7, P14, P21, P28, and P120.

## RiboTag RNAseq

All mice were collected between 9:30 am and 12:30 pm on the day of experiment. Mice were anesthetized by I.P. injection of 100 mg/kg ketamine (Victor Medical Company)/20 mg/kg xylazine (Anased) mix, and transcardially perfused with 10 ml PBS then 10 ml 1% PFA. Brains were dissected in 2.5 mM HEPES-KOH pH 7.4, 35 mM glucose, 4 mM NaHCO3 in 1× Hank's Balanced Salt Solution with 100 µg/ml cycloheximide added fresh (*Heiman et al., 2014*). Brains were cut at approximately bregma –2.4 to isolate the VC, the cortex was carefully detached from the subcortical areas, and any visible white matter was removed. Lateral cuts were made at 1 mm and 3 mm from the midline to further isolate the VC section, and RiboTag pulldown was immediately performed. For each time point, the visual cortices from two mice (Rpl22-HA+; Gfap cre+) were pooled for RNA isolation and RNA sequencing library preparation. P7 = 3 biological replicates (6 mice, 2 × 3); P14 = 4 biological replicates (8 mice, 2 × 4); P28 = 5 biological replicates (10 mice, 2 × 5); P120 = 6 biological replicates, 3 new samples (6 mice, 2 × 3), plus for data analysis 3 additional P120 biological replicates from a previously published study from the lab (*Boisvert et al., 2018*; GEO GSE99791), collected and processed in the same way, were included to increase the power of the analysis.

## Histology (smFISH in situ hybridization and immunostaining)

Mice aged P4 and older were anesthetized by I.P. injection of 100 mg/kg ketamine (Victor Medical Company)/20 mg/kg xylazine (Anased) mix and transcardially perfused with PBS, then 4% PFA at room temperature. Brains were removed and incubated in 4% PFA overnight at 4 °C, then washed 3 × 5 min with PBS, and cryoprotected in 30% sucrose for 2–3 days, before being embedded in TFM media (General Data Healthcare #TFM-5), frozen in dry ice-ethanol slurry solution, and stored at –80° C until use. P1 mice were decapitated and brains removed without perfusion, briefly washed in PBS and placed in 4% PFA overnight at 4° C, followed by a similar procedure as described above for older mice. Brains were sectioned using a cryostat (Hacker Industries #OTF5000) in sagittal or coronal orientations depending on experimental needs at a slice thickness of 16–25 μm. Sections were mounted on Superfrost Plus slides (Fisher #1255015). Immunostaining for synaptic markers and smFISH was performed on the same day of sectioning. 3–5 mice were used for each experimental group. For each mouse, three sections were imaged and analyzed.

## Single-nucleus RNAseq and western blot

Mice were anesthetized by I.P. injection of 100 mg/kg ketamine (Victor Medical Company)/20 mg/kg xylazine (Anased) mix, then decapitated. Brains were rapidly removed and the VC dissected in ice-cold PBS using the same coordinates as described for RiboTag RNAseq. Dissected cortices were snap frozen, and kept at –80° C until use. For snRNAseq, 4 mice were collected for each experimental group. For western blot, 2–4 independent experiments/samples for each condition were analyzed.

## RNAseq
### Bulk RNAseq using RiboTag
#### RiboTag pulldown

A modified RiboTag protocol was performed to isolate astrocyte-enriched RNA. Briefly, brain samples were homogenized using a Dounce homogenizer (Sigma #D9063) in 2 ml cycloheximide-supplemented homogenization buffer (1% NP-40, 0.1 M KCl, 0.05 M Tris, pH 7.4, 0.012 M $MgCl_2$ in RNase-free water, with 1:1000 1 M DTT, 1 mg/ml heparin, 0.1 mg/ml cycloheximide, 1:100 protease inhibitors, and 1:200 RNAsin added fresh). Homogenates were centrifuged and the supernatant incubated on a rotator at 4°C for 4 hr with 5 μμl anti-HA antibody to bind the HA-tagged ribosomes (CST Rb anti-HA #3724, 1:200). Magnetic IgG beads (Thermo Scientific Pierce #88847) were conjugated to the antibody-ribosome complex via overnight incubation on a rotator at 4°C. Samples were washed with a high salt buffer (0.333 M KCl, 1% NP40, 1:2000 1 M DTT, 0.1 mg/ml cycloheximide, 0.05 M Tris pH 7.4, 0.012 M $MgCl_2$ in RNase-free water), and RNA released from ribosomes with 350 μl RLT buffer (from Qiagen RNeasy kit) with 1% BME. RNA was purified using RNeasy Plus Micro kit (Qiagen 74034) according to the manufacturer's instructions and eluted into 16 μl RNase-free water. Eluted RNA was stored at –80 °C. For each time point, 50 μl of homogenate (pre-anti-HA antibody addition) was set aside after centrifugation, kept at –20 °C overnight, and purified via RNeasy Micro kit as an 'input' sample, and used to determine astrocyte enrichment.

#### Library generation and sequencing

RNA quantity and quality were measured with a Tape Station (Agilent) and Qubit Fluorimeter (Thermo Fisher) before library preparation. >100 ng of RNA was used to make libraries. mRNA was extracted with oligo-dT beads, capturing polyA tails, and cDNA libraries made with Illumina TruSeq Stranded mRNA Library Preparation Kit (RS-122-2101) by the Salk Institute Next Generation Sequencing (NGS) Core. Samples were sequenced on an Illumina HiSeq 2500 with single-end 50 base-pair reads, at 12–70 million reads per sample.

#### RNA sequencing mapping, analysis, and statistics

Raw sequencing data was demultiplexed and converted into FASTQ files using CASAVA (v1.8.2) and quality tested with FASTQC v0.11.2. Alignment to the mm10 genome was performed using the STAR aligner version 2.5.1b (*Dobin et al., 2013*). Mapping was carried out using default parameters (up to 10 mismatches per read, and up to 9 multi-mapping locations per read), and a high ratio of uniquely mapped reads (>75%) was confirmed with exonic alignment inspected to ensure that reads were mapped predominantly to annotated exons. Raw and normalized (FPKM) gene expression was

quantified across all genes (RNAseq) using the top-expressed isoform as a proxy for gene expression using HOMER v4.10 (*Heinz et al., 2010*), resulting in 10–55 million uniquely mapped reads in exons. Principal component analysis was carried out with prcomp in R 3.4.3 on normalized counts. Differential gene expression was carried out using the DESeq2 (*Love et al., 2014*) package version 1.14.1 using the HOMER getDiffExpression.pl script with default normalization and using replicates to compute within-group dispersion. Significance for differential expression was defined as adjusted p<0.05 (also labeled as FDR), calculated using Benjamini–Hochberg's procedure for multiple comparison adjustment.

Significantly altered genes are presented in three categories:

*All genes*: FPKM >1, adjusted p<0.05.

*Astrocyte-expressed genes*: RiboTag pulldown (astrocyte)/input (all cells) > 0.75, FPKM > 1, adjusted p<0.05.

*Astrocyte-enriched genes*: RiboTag pulldown (astrocyte)/input (all cells) > 3, FPKM > 1, adjusted p<0.05.

See also *Boisvert et al., 2018*. A full list of genes in each time point is presented in *Figure 1—source data 1A*.

## GO enrichment analysis

GO terms that are enriched in astrocytes at each developmental stage were identified using the String database (https://string-db.org/)(*Szklarczyk et al., 2019*). A search using multiple proteins by gene name was performed with the default parameters, and GO BP category selected and exported from the analysis tab. GO terms with gene ratio of 0.5 and above were selected, and plotted for each age group, with x-axis showing the ratio of genes overlapping with each GO term, and bar fill color is the significance of the overlap (adj. p-value; FDR). GO terms common to all age groups were obtained using the Venn diagram (http://www.interactivenn.net/; *Heberle et al., 2015*), and terms with gene ratio equal to or above 0.5 were selected and plotted. A full list of GO terms is presented in *Figure 1—source data 1B*.

## Single-nucleus RNAseq

### Sample preparation

A total of 8 samples (two for VGlut2 WT, two for VGlut2 cKO, two for Ip3r2 WT, two for Ip3r2 KO) were sequenced to obtain the dataset described in *Figure 6*, *Figure 6—figure supplement 1*, *Figure 6—figure supplement 2*. The samples were as follows: VGlut2 WT_1; VGlut2 cKO_1; VGlut2 WT_2; VGlut2 cKO_2; Ip3r2 WT_1; Ip3r2 KO_1; Ip3r2 WT_2; Ip3r2 KO_2. Each group consisted of one replicate from male mice and one replicate from female mice. Each replicate consisted of the VC from both hemispheres of two mice of the same genotype and gender. Nuclear isolation, FACS sorting, 10× barcoding, and cDNA preparation were performed on the same day using one WT and KO pair, which were processed in parallel, resulting in four separate procedures. cDNA was stored at –20°C until all samples were collected. Library preparation and sequencing were carried out at the same time for all eight samples.

### Nuclei preparation

Nuclei were isolated from frozen VC tissue. Tissue was manually homogenized using a two-step Dounce homogenizer (A and B) (Sigma #D9063) in NIMT buffer, containing (in mM: 250 sucrose, 25 KCl, 5 MgCl$_2$, 10 Tris-Cl pH 8, 1 DTT; 1:100 dilution of Triton X100, Protease Inhibitor Cocktail [Sigma #P8340]; and 1:1000 dilution of RNaseOUT Recombinant Ribonuclease Inhibitor [Thermo #10777019]; SUPERase• In RNase Inhibitor [Thermo #AM2694]) on ice. Homogenized samples were mixed with 50% iodixanol (OptiPrep Density Gradient Medium; Sigma #D1556) and loaded onto 25% iodixanol cushion, and centrifuged at 10,000 g for 20 min at 4°C in a swinging bucket rotor (Sorval HS-4). Pellets resuspended in ice-cold DPBS (HyClone) with 1:1000 dilution of RNaseOUT Recombinant Ribonuclease Inhibitor (Thermo #10777019); SUPERase• In RNase Inhibitor (Thermo #AM2694). Nuclei were then incubated for 7 min on ice with Hoechst 33342 solution (20 mM) (Thermo #62249) (final concentration 0.5 µM), followed by centrifugation at 1000 g for 10 min at 4° C to pellet nuclei. Pellets were

resuspended in blocking buffer containing DPBS with RNAse inhibitors, and 1:10 dilution of pure BSA, and blocked for 30 min on ice. NEUN-Alexa488 pre-conjugated antibody (Millipore #MAB377X) was then added at 1:1000 dilution and incubated for at least 1 hr on ice before proceeding to flow cytometry sorting.

## Flow cytometry

Fluorescence-activated nuclei sorting (FANS) was performed in the Salk Institute Flow Cytometry core using a BD FACS Aria Fusion sorter with PBS for sheath fluid (a 100 µm nozzle was used for these experiments with sheath pressure set to 20 PSI). Hoechst-positive nuclei were gated first (fluorescence measured in the BV421 channel), followed by exclusion of debris using forward and side scatter pulse area parameters (FSC-A and SSC-A), exclusion of aggregates using pulse width (FSC-W and SSC-W), before gating populations based on NEUN fluorescence (using the FITC channel). To isolate the non-neuronal cell population, nuclei devoid of FITC signal (NEUN-) were collected (*Figure 6A*). Nuclei were purified using a one-drop single-cell sort mode (for counting accuracy); these were directly deposited into a 1.5 ml Eppendorf without additional buffer (to yield a sufficient concentration that permitted direct loading onto the 10× chip).

Sorted NeuN-negative nuclei were immediately processed with 10X Chromium kit (10X Genomics) for single-nucleus barcoding. Nuclei were kept on ice for the entire process. At each time, WT and KO samples were processed in parallel on the same day.

## 10X Chromium barcoding, library preparation, and sequencing

Single-nuclei separation, barcoding, and cDNA generation were performed following the manufacturer's instruction using the Chromium single cell 3′ kit (V3, 10X Genomics PN-1000073). cDNA concentration and quality were measured using Qubit Fluorimeter (Thermo Fisher) and Tape Station (Agilent), respectively, and was stored at –20° C until library preparation.

Libraries were generated from all samples at the same time (eight total samples, 2 WT/2 KO Vglut2 cKO model; 2 WT/2KO IP3R2 KO model) following the manufacturer's instructions using the Chromium single cell 3′ kit (V3, 10X Genomics PN-1000075). Library quality was assessed with a Tape station (Agilent). NovaSeq sequencing was performed at the UCSF Center for Advanced Technology, at ~300 million reads/sample (60,000 reads/cell).

## Single-cell RNAseq data preprocessing and clustering

Data was demultiplexed and mapped onto the mouse genome (mm10) using 10X Cellranger (v3.1.0) with default parameters. Cell barcodes with <200 genes detected were discarded due to low coverage. Doublets were identified and removed using Scrublet (*Wolock et al., 2019*) with its default setting in each sample. The average number of UMIs per cell was 2310 ± 878; average number of genes detected per cell (UMI ≥ 1) was 1168 ± 328. Cell clusters were identified using Scanpy (v1.4.3), following the clustering process described in *Luecken and Theis, 2019*. All the samples were combined and used the top 5000 highly variable genes as the input dimension reduction. To identify clusters, Scanorama (v1.0.0, default parameter, k = 20; *Hie et al., 2019*) was used to perform batch correction and dimension reduction (30 PCs), followed by Leiden clustering (*Traag et al., 2019*; resolution = 1). Data was visualized using the UMAP embedding (*McInnes et al., 2018*) function from Scanpy. The ensemble clustering identified all astrocytes as one cluster, and to further identify astrocytes subtypes, we repeated the same clustering process on the astrocytes cluster only and got four subtypes. Astrocyte clusters were annotated using cell-type marker genes identified from previous studies to label distinct cortical astrocyte populations (*Bayraktar et al., 2020*; *Lanjakornsiripan et al., 2018*; *Marques et al., 2016*; *Tasic et al., 2018*; *Van Hove et al., 2019*; *Zeisel et al., 2018*). A full list of genes in each layer group is presented in *Figure 6—source data 1A and B*.

## Identifying DEGs

To identify cluster-specific DEGs, we used the scanpy.tl.rank_gene_groups function to perform the Wilcoxon rank-sum test with Benjamini–Hochberg correction to compare cells from each cluster with the remaining cells. Genes with FDR < 0.1 and log2 FC between –0.15 and 0.15 were identified as DEGs. To identify DEGs between KO and WT, we performed the same analysis using combined

astrocyte clusters. All comparisons were performed separately for VGlut2 cKO and Ip3r2 KO samples. A full list of DEGs is presented in *Figure 6—source data 1D*.

## GO enrichment analysis

GO terms that are enriched in astrocyte gene groups within each cluster, as well as genes regulated by neuronal or astrocyte activity, were identified using the String database (https://string-db.org/)( *Szklarczyk et al., 2019*). A search using multiple proteins by gene name was performed separately on VGlut2 cKO and Ip3r2 KO samples, and up- and downregulated DEGs, using the default parameters, and GO BP category selected and exported from the "analysis" tab. 20 GO terms with highest gene ratio were selected and plotted for each model, with x-axis showing the ratio of genes overlapping with each GO term, and bar fill color is significance of the overlap (adj. p-value; FDR). GO terms common to both models were obtained using the Venn diagram (http://www.interactivenn.net/; *Heberle et al., 2015*) and plotted. A full list of GO terms is presented in *Figure 6—source data 1C and G*.

## Cell culture

### RGC neuron purification and culture

RGC purification and culture was performed as described (*Allen et al., 2012*; *Ullian et al., 2001*; *Winzeler and Wang, 2013*). Briefly, retinas from P5 to P7 rat pups of both sexes were removed and placed in DPBS (HyClone #SH30264). Retinas were digested with Papain (Worthington #PAP2 3176; 50 units) for 30 min at 34°C , triturated with low OVO (15 mg/ml trypsin inhibitor; Worthington #LS003086), then high OVO (30 mg/ml trypsin inhibitor; Worthington #LS003086) solutions. The cell suspension was then added to lectin (Vector #L-1100)-coated Petri dishes to pull down microglia and fibroblast cells for 5–10 min at room temperature. The remaining cells were then added to T11D7 hybridoma supernatant-coated Petri dishes for 40 min at room temperature, which specifically binds RGCs. After washing off the non-binding cells with DPBS, pure RGCs were released by trypsin treatment (Sigma #T9935) to cleave cell-antibody bond and collected. RGCs were plated on six-well plates coated with PDL (Sigma # P6407) and laminin (Cultrex Trevigen #3400-010-01) at a density of 125,000 cells/well. RGCs were maintained in the following media: 50% DMEM (LifeTech #11960044); 50% Neurobasal (LifeTech #21103049); Penicillin-Streptomycin (LifeTech #15140-122); glutamax (LifeTech #35050-061); sodium pyruvate (LifeTech #11360-070); N-acetyl-L-cysteine (NAC) (Sigma #A8199); insulin (Sigma #I1882); triiodo-thyronine (Sigma #T6397); SATO (containing transferrin [Sigma #T-1147]; BSA [Sigma #A-4161]; progesterone [Sigma #P6149]; putrescine [Sigma #P5780]; sodium selenite [Sigma #S9133]); and B27 (see *Winzeler and Wang, 2013* for recipe). For complete growth media, the media was supplemented with BDNF (PeproTech #450-02), CNTF (PeproTech #450-13), and forskolin (Sigma #F6886). The next day, half of the media was replaced with media containing FUDR (13 µg/µl final concentration; Sigma #F0503) to inhibit fibroblast growth. Cells were fed by replacing half of the media with fresh equilibrated media every 3–4 days. RGCs were maintained at 37° C/10% $CO_2$ and kept in culture for at least 7 days prior to treatment to allow for full process outgrowth.

### Astrocyte preparation and culture

Primary astrocytes from rat cortex were prepared as described (*Allen et al., 2012*; *McCarthy and de Vellis, 1980*). Briefly, the cerebral cortex from P1 to P2 rat pups were removed and placed in DPBS (HyClone #SH30264). The meninges and hippocampi were removed and discarded. The remaining cortices were diced and digested with Papain (Worthington #LS003126; 330 units) for 1 hr and 15 min in 37° C 10% $CO_2$ cell culture incubator. Cells were triturated in low OVO and then high OVO-containing solutions, and plated in PDL-coated 75 cm tissue culture flasks. 3 days after plating, flasks were manually shaken to remove upper cell layers, which contained mostly non-astrocytic cells. 2 days after shake off, ARA-C (10 µM final concentration; Sigma #C1768) was added for 48 hr to inhibit the other proliferating cells, which divide faster than astrocytes. Finally, astrocytes were plated in 15 cm cell culture plates coated with PDL at 2–3 million cells/dish and passaged once a week. Astrocytes were maintained at 37° C/10% $CO_2$ and kept in culture for 3–4 weeks. Astrocyte culture medium was DMEM (LifeTech #11960044) supplemented with 10% heat-inactivated FBS (LifeTech #10437028), Penicillin-Streptomycin (LifeTech #15140-122), glutamax (LifeTech #35050-061), Insulin (Sigma

#I1882), sodium pyruvate (LifeTech #11360-070), hydrocortisone (Sigma #H0888), and N-acetyl-L-cysteine (Sigma #A8199).

## Treatment of astrocyte cultures with cultured neurons

Cultured astrocytes were plated on cell culture inserts (Falcon #353102) at 250,000 cells/insert. Inserts were added to six-well plates containing either plated RGC neurons (at ~125,000 cells/well) or empty wells coated with PDL and laminin (similar to RGC-plated wells) and containing media. Cells were incubated together for 4 days in low protein conditioning media containing (50% DMEM, 50% Neurobasal media; Penicillin-Streptomycin; glutamax and sodium pyruvate, NAC, BDNF, CNTF, forskolin), after which conditioned media was collected and concentrated 50-fold using 10 kDa cutoff concentrators (Sartorius #14558502). Protein concentration was measured using the Bradford assay. Three experimental groups were compared: RGCs alone, astrocytes alone, and astrocytes + RGCs.

## Treatment of astrocyte cultures with neurotransmitters

Astrocytes were plated on six-well plates at 150,000 cell/well and allowed to reach 90% confluency (1–2 days). Then astrocytes were incubated for 48 hr in low protein medium containing (50% DMEM, 50% Neurobasal media; Penicillin-Streptomycin; glutamax and sodium pyruvate) alone (control), or with either 100 µM glutamate (Sigma #G5889-100G), 10 µM adenosine (Sigma #A4036-5G), or 100 µM ATP (Sigma #A6419-1G) final concentration. Conditioned media was then collected as described above. Concentration was measured with Bradford assay. Samples were stored at 4° C for up to 7 days or processed immediately for western blot.

## Western blot

Samples were heated in reducing loading dye (Thermo # 39000) for 45 min at 55° C. For conditioned media, 10 µg/well was loaded; for tissue lysates, 20 µg/lane was loaded. Samples were resolved on 4–12% bis-tris or bolt gels (Invitrogen #NW04120) for 30–40 min at 150–200 V. Proteins were transferred to PVDF membranes at 100 V for 1 hr, then blocked in 1 % casein (Bio-Rad #1610782) in TBS (Bioworld #105300272) blocking buffer for 1 hr at room temperature on a shaker. Primary antibodies were applied overnight at 4° C diluted in blocking buffer. The antibodies used were Rb anti-glypican 4 (Proteintech #13048-1-AP; 1:500), Rb anti-IP3R2 (a gift from Ju Chen lab, UCSD 1:1000), and Ms anti-tubulin (Thermo #MA5-16308 1:5000). The next day, membranes were washed 3 × 10 min with TBS-0.1% Tween and the appropriate secondary antibody conjugated to Alexa Fluor 680 (Molecular Probes) was applied for 2 hr at room temperature (dilution 1:10,000). Bands were visualized using the Odyssey Infrared Imager (LI-COR) and band intensity analyzed using the Image Studio software (LI-COR).

## Histology

### Immunostaining in mouse brain tissue

The slides containing the sections were blocked for 1 hr at room temperature in blocking buffer containing antibody buffer (100 mM L-lysine and 0.3% Triton X-100 in PBS) supplemented with 10% heat-inactivated normal goat serum. Primary antibodies diluted in antibody buffer with 5% goat serum were incubated overnight at 4° C. The next day, slides were washed 3 × 5 min with PBS with 0.2% Triton X-100 and secondary antibodies conjugated to Alexa Fluor (Molecular Probes) were applied for 2 hr at room temperature. Slides were mounted with the SlowFade Gold with DAPI mounting media (LifeTech #S36939), covered with 1.5 glass coverslip (Fisher #12544E) and sealed with clear nail polish. The following antibodies were used: Chk anti-GFP (Millipore #06-896, 1:500), Rb anti-SOX9 (Abcam #ab185966, 1:2000), Rb anti-ALDH1L1 (Abcam #ab-87117, 1:500), Rb anti-HA (CST #3724), Rb anti-S100β (Abcam #ab52642, 1:100), Ms anti-NEUN (Millipore #MAB377 1:100), Rb anti-NG2 (Millipore # Ab5320), Rb anti-MOG (Proteintech # 12690-1-ap), Rb anti-IBA1 (Wako #016-20001), Gp anti-VGLUT1 (Millipore #AB5905, 1:2000), Gp anti-VGLUT2 (Millipore #AB2251 1:3000, 1:5000), Rb anti-GLUA1 (Millipore #AB1504, 1:400), Rb anti-GLUA2 (Millipore #AB1768-I, 1:400), and Ms anti-Bassoon (Enzo #VAMP500, 1:500). All secondary antibodies were applied at 1:500 dilution.

The following mouse lines and antibody combinations were used:

| Experiment and figure # | Antibody targets |
|---|---|
| Cell marker colocalization RiboTag validation (*Figure 1*, *Figure 1—figure supplement 1*) | HA, S100β, NEUN, IBA1, NG2, MOG |
| Astrocyte number across development per layer (*Figure 2*, *Figure 2—figure supplement 1*) | GFP, VGLUT2 |
| Astrocyte marker colocalization with Aldh1l1-Gfp (*Figure 2—figure supplement 1*) | GFP, SOX9, S100β, ALDH1L1 |
| Presynaptic development per layer (*Figure 3*, *Figure 3—figure supplement 1*) | GFP, VGLUT1, VGLUT2 |
| Postsynaptic development per layer (*Figure 3*) | GFP, GLUA1, GLUA2 |
| Assessing the presence of thalamic projections to the VC in VGlut2 cKO mice (*Figure 4—figure supplement 2*) | VGLUT2 |
| Analysis of VGLUT1 and VGLUT2 signal in VGlut2 cKO mice (*Figure 4*, *Figure 4—figure supplement 2*) | VGLUT1, VGLUT2 |
| Quantification of pre and postsynaptic puncta, and synapses (*Figure 4*, *Figure 4—figure supplement 2*, *Figure 5*, *Figure 5—figure supplement 1*) | VGLUT1, VGLUT2, Bassoon, GLUA1, GLUA2 |

## Single-molecule fluorescent in situ hybridization (smFISH)

All smFISH experiments reported here were performed on brain tissue fixed with 4% paraformaldehyde and processed using cryosectioning as described in the 'Mouse tissue collection' section. The assay was performed using the RNAscope 2.5 HD—multiplex fluorescent Manual Assay kit (ACDbio #320850) using the manufacturer's instructions for fixed-frozen tissue with the following modifications. Directly following cryosectioning, slides containing brain sections were dried for 1 hr at –20° C, then washed for 5 min in PBS at room temperature, followed by brief wash (~1 min) in 100% Molecular Biology Grade Ethanol. The slides were then air-dried for 5 min and incubated with appropriate pretreatment reagents at 40° C. For tissue from P1 to P7 mice, slides were incubated with protease 3 for 30 min; for P14–P28, protease 4 30 min. Slides were then briefly washed with PBS and incubated with target probes for 2 hr at 40° C, followed by three amplification steps and one detection step. Slides were mounted using the SlowFade Gold with DAPI mounting media (LifeTech #S36939) covered with 1.5 glass coverslip (Fisher #12544E) and sealed with clear nail polish. The original protocol can be found in ACDbio website: https://acdbio.com/technical-support/user-manuals. Detailed step-by-step modified protocol performed here is available upon request.

All slides were either imaged within 1–2 days or stored at –20° C until imaging.

## Imaging and analysis

### Fluorescent microscopy

Imaging was performed using an Axio Imager.Z2 fluorescent microscope (Zeiss) with the apotome module (apotome 2.0) and AxioCam HR3 camera (Zeiss) at 20× magnification. Tile images that contain the entire primary VC (from pial surface to white matter tract) were acquired. Number of tiles adjusted to contain a similar area of the cortex at each developmental stage, typically 1–2 (width) × 2–4 (depth) (pixel size 0.3 × 0.3 µm).

For RiboTag validation (*Figure 1*, *Figure 1—figure supplement 1*): Single-plane images were obtained.

For in situ hybridization experiments (*Figure 2*, *Figure 2—figure supplement 1*): z-stack images (seven slices, optical slice 1 µm) were obtained.

For developmental analysis of astrocyte numbers per layer (*Figure 2*, *Figure 2—figure supplement 1*), presynaptic marker analysis during development (*Figure 3*, *Figure 3—figure supplement 1*), VGlut2 cKO validation (*Figure 4*, *Figure 4—figure supplement 2*), and Aldh1l1-GFP mouse validation (*Figure 2—figure supplement 1*): z stack images (three slices, optical slice 1 µm) were obtained.

### Confocal microscopy

*Developmental analysis of GLUA1 and GLUA2 expression (Figure 3)*: Slides were imaged using Zeiss LSM 700 confocal microscope at 63× magnification. A 1176 × 1176 pixel 2.7 µm thick z-stack image

was obtained (pixel size 0.09 × 0.09 × 0.3 µm, 10 slices per 2.7 µm stack). In total, four images were taken from each section to encompass all cortical layers. Layers 4–5 were combined into one image.

*RORacre-tdTomato+ thalamic projections in the VC* (*Figure 4—figure supplement 2*): Slides were imaged using Zeiss LSM 700 confocal microscope at 63× magnification. A 900 × 900 pixels 2.7 µm thick z-stack image was obtained (xyz size 0.11 × 0.11 × 0.3 µm, 10 slices per 2.7 µm stack). Separate images were taken for layer 1 and layer 4.

*Synapse number analysis* (*Figure 4*, *Figure 4—figure supplement 2*, *Figure 5*, *Figure 5—figure supplement 1*): Slides were imaged using Zeiss LSM 880 confocal microscope at 63× magnification. For each section, 1420 × 920 pixels 3.5 µm thick z-stack image was obtained (pixel size 0.08 × 0.08 × 0.39 µm; 10 slices per 3.5 µm stack). All images were from layer 1. Example images show a single z plane from the same location in the stack for both genotypes.

In all cases, when comparing WT and KO per given experiment, slides were imaged on the same day using set exposure.

## Image analysis

Image analysis was primarily done with ImageJ (FIJI, NIH) or Imaris (Bitplane) software as described below for each section:

*Cell marker colocalization RiboTag validation* (*Figure 1*, *Figure 1—figure supplement 1*): This was performed using FIJI (ImageJ). Thresholding was performed on the RiboTag-labeled image (stained with an anti-HA tag antibody), and the 'Analyze Particles' function was used with a minimum area of 20–40 µm to automatically separate and quantify the total number of RiboTag-positive cells. The number of double-labeled RiboTag and cell-type antibody-positive cells were manually counted. This generated the proportion of RiboTag-positive cells that also label for the cell-specific marker.

*Astrocyte number across development per layer* (*Figure 2*, *Figure 2—figure supplement 1*): This was done on sections of Aldh1l1-GFP VC that were co-immunostained for VGLUT2 using semi-automatic custom-made macro in ImageJ. For each image, maximal intensity projections were created, then each cortical layer was manually cropped based on DAPI and VGLUT2 staining, and saved as a separate file. Then, a colocalization file was created using the 'colocalization threshold' function to merge the colocalized cell marked by DAPI with the Aldh1l1-GFP signal to specifically select astrocytes. Colocalized objects were counted using the 'Analyze Particles' function. Number of astrocytes in each layer was recorded for each developmental stage.

The high cell density at early ages made it impossible to use a similar method to count the total number of cells using the DAPI labels. Instead, three Regions of interest (ROIs) were created for each layer, and cells were counted manually within each ROI, using the 'multi-point' tool in ImageJ. The total cell number in each layer was then extrapolated based on the total area measurement in each file.

*Astrocyte marker colocalization with Aldh1l1-Gfp* (*Figure 2—figure supplement 1*): An ROI containing the entire depth of the cortex of a maximal intensity projection image was cropped equally for each image. Labeled cells were counted manually using the 'cell counter' plugin in ImageJ. Positively labeled cells were identified based on signal strength. For each file, three types of counts were made: the appropriate astrocyte marker-positive cell number, Aldh1l1-GFP-positive cell number, and colocalized cell number.

*Spatio-temporal analysis of presynaptic proteins* (*Figure 3*, *Figure 3—figure supplement 1*): This was performed with ImageJ. As above, for each file, maximal intensity projections were created, then layers were cropped out manually and saved as separate files. VGLUT signal was thresholded in the same way for all images to contain all visible signal. The threshold area measurement was recorded for each file.

*Spatio-temporal analysis of postsynaptic proteins* (*Figure 3*): This was performed using the Imaris software (Bitplane). GLUA puncta number was calculated using the 'spots' function, and mean intensity filter to select the positive puncta. In all images, signals were thresholded in the same way. To analyze specifically the GLUA signal in the cell processes and not the soma, cell bodies labeled by DAPI were selected manually using the 'create object' function. Then GLUA puncta number that colocalized with cell bodies was quantified. Finally, cell body-related GLUA1 puncta were subtracted from the total puncta number to obtain process-expressed GLUAs.

*Quantification of tdTomato+ thalamocortical projections in the VC* (***Figure 4—figure supplement 2***): was performed using Imaris (Bitplane). tdTomato+ processes were rendered using the 'create object' function. In all images, signals were thresholded in the same way to select labeled processes. Total volume was calculated and compared between the experimental groups. VGLUT2 puncta were rendered using the 'create spots' function, and mean intensity filter to threshold positive spots.

*Analysis of VGLUT1 and VGLUT2 signal in VGlut2 cKO mice* (***Figure 4***, ***Figure 4—figure supplement 2***): was performed using ImageJ as described for developmental presynaptic experiments.

*Counting astrocytes within each layer in VGlut2 cKO and Ip3r2 KO mice* (***Figure 4—figure supplement 2***, ***Figure 5—figure supplement 1***): was performed manually using the 'cell counter' plugin in ImageJ using smFISH in situ images (see below). Astrocytes were identified by positive *Slc1a3* probe signal.

*Quantification of smFISH signal* (***Figure 2***, ***Figure 2—figure supplement 2***, ***Figure 4***, ***Figure 4—figure supplement 1***, ***Figure 5***, ***Figure 5—figure supplement 1***): was performed using a custom-made macro in ImageJ. Maximal intensity projection images of the VC were manually cropped per layer and saved as individual files. Astrocytes were identified using the GFP signal in experiments with Aldh1l1-GFP mice (***Figure 2***, ***Figure 2—figure supplement 2***); and *Slc1a3* probe signal in all other experiments (***Figure 4***, ***Figure 4—figure supplement 2***, ***Figure 5***, ***Figure 5—figure supplement 1***). An ROI outline was then created around the cell body of the astrocyte, the signal of the probe of interest was then thresholded in the same way for all images, and the threshold area was recorded for each cell. This strategy was employed because we found that threshold area of the signal gave a more reliable and stable result than intensity measurements, which are not recommended by smFISH protocol (RNAscope by ACDbio). Moreover, due to the density of the signal in some cases, counting individual puncta was impossible.

*Quantification of pre-, postsynaptic puncta, and synapses* (***Figure 4***, ***Figure 4—figure supplement 2***, ***Figure 5***, ***Figure 5—figure supplement 1***): This was performed using Imaris software (Bitplane). Positive puncta of GLUA1, GLUA2, VGLUT1, VGLUT2, Bassoon, and tdTomato processes (***Figure 4—figure supplement 2***) were selected by size and intensity by thresholding the signal in the same way for each section. Then colocalization between each two pre-postsynaptic pairs was calculated. Puncta were considered colocalized if the distance between them was ≤0.5 µm (***Blanco-Suarez et al., 2018***; ***Farhy-Tselnicker et al., 2017***). For experiments described in ***Figure 4—figure supplement 2***, first colocalization between tdTomato and Bassoon was established and cropped. The colocalized Bassoon-tdTomato puncta were then used to calculate colocalization with GLUA1 or GLUA2. Number of colocalized puncta was obtained and compared between the experimental groups. A minimum of three sections per mouse were imaged for each brain region, and the experiment was repeated in at least five WT and KO pairs. Example images show a single z plane from the same location in the stack for both genotypes.

## Data presentation and statistical analysis

All data is presented as either mean ± s.e.m., scatter with mean ± s.e.m., or scatter with range, as indicated in each figure legend. Statistical analysis was performed using Prism software (GraphPad) unless otherwise stated in the text. Multiple group comparisons were done using one-way analysis of variance (ANOVA) with post hoc Tukey's or Dunn's tests. Pairwise comparisons were done by t-test. When data did not pass the normal distribution test, multiple comparisons were done by Kruskal–Wallis ANOVA on ranks and pairwise comparisons were done with the Mann–Whitney rank-sum test. p-Value ≤ 0.05 was considered statistically significant. Analysis was done blind to genotype. The sample sizes, statistical tests used, and significance are presented in each figure and figure legend.

## Acknowledgements

We thank members of the Allen lab for helpful discussions on the project. This work was supported by NIH NINDS grants to NJA: NS105742 and NS089791. Work in the lab of NJA was supported by the Hearst Foundation, the Pew Foundation and the CZI Neurodegeneration Network. This work was supported by Core Facilities of the Salk Institute (Next Generation Sequencing, Bioinformatics, Biophotonics: NIH NCI CCSG P30 014195, RO1 GM102491-06, R01AG064049-01, P30AG068635, P01 AG073084-01 the Waitt, Helmsley and Chapman Foundations). JRE is an Investigator of the Howard Hughes Medical Institute.

# Additional information

## Funding

| Funder | Grant reference number | Author |
|---|---|---|
| National Institutes of Health | NS105742 | Nicola J Allen |
| National Institutes of Health | NS089791 | Nicola J Allen |
| Pew Charitable Trusts | | Nicola J Allen |
| Chan Zuckerberg Initiative | 2018-191894 | Nicola J Allen |
| Howard Hughes Medical Institute | | Joseph R Ecker |
| National Institutes of Health | NIH NCI CCSG P30 014195 | Maxim N Shokhirev |
| Hearst Foundations | | Nicola J Allen |
| National Institutes of Health | GM102491-06 | Maxim N Shokhirev |
| National Institutes of Health | AG064049-01 | Maxim N Shokhirev |
| National Institutes of Health | P30AG068635 | Maxim N Shokhirev |
| National Institutes of Health | AG073084-01 | Maxim N Shokhirev |

The funders had no role in study design, data collection and interpretation, or the decision to submit the work for publication.

## Author contributions

Isabella Farhy-Tselnicker, Conceptualization, Formal analysis, Investigation, Writing – original draft, Supervision, Visualization, Writing – review and editing; Matthew M Boisvert, Elena Blanco-Suarez, Formal analysis, Investigation, Writing – review and editing; Hanqing Liu, Galina A Erikson, Chen Farhy, Formal analysis, Writing – review and editing; Cari Dowling, Investigation, Writing – review and editing; Maxim N Shokhirev, Formal analysis, Supervision, Writing – review and editing; Joseph R Ecker, Supervision, Writing – review and editing; Nicola J Allen, Conceptualization, Formal analysis, Funding acquisition, Supervision, Writing – original draft

## Author ORCIDs

Isabella Farhy-Tselnicker 
Elena Blanco-Suarez 
Chen Farhy 
Maxim N Shokhirev 
Joseph R Ecker 
Nicola J Allen 

## Ethics

This study was performed in accordance with the Guide for the Care and Use of Laboratory Animals of the National Institutes of Health. All animal work was approved by the Salk Institute Institutional Animal Care and Use Committee (IACUC) protocol 12-00023.

## Decision letter and Author response

Decision letter https://doi.org/10.7554/eLife.70514.sa1
Author response https://doi.org/10.7554/eLife.70514.sa2

## Additional files

### Supplementary files
• Transparent reporting form

### Data availability
All data supporting the results of this study can be found in the following locations: Processed RNA sequencing data presented in Figure 1 is available in Figure 1- Source Data 1. Data presented in graphs in Figures 2-5 is available in Figure 2-5 Source Data files. Processed single nucleus RNA sequencing data presented in Figure 6 is available in Figure 6-Source Data 1. The RNA sequencing data has been deposited at GEO. Ribotag data is available at GSE161398 and glial snRNAseq at GSE163775. Processed RNA sequencing data is available in a searchable format at the following web locations: ribotag astrocyte developmental dataset: http://igc1.salk.edu:3838/astrocyte_transcriptome/; single nucleus glial cell sequencing: http://mouse-astro-dev.cells.ucsc.edu/.

The following dataset was generated:

| Author(s) | Year | Dataset title | Dataset URL | Database and Identifier |
|---|---|---|---|---|
| Boisvert MM, Farhy-Tselnicker I, Erikson GA, Shokhirev MN, Allen NJ | 2021 | Astrocyte developmental transcriptome from mouse visual cortex | https://www.ncbi.nlm.nih.gov/geo/query/acc.cgi?acc=GSE161398 | NCBI Gene Expression Omnibus, GSE161398 |
| Farhy-Tselnicker I, Liu H, Shokhirev MN, Ecker JR, Allen NJ | 2021 | Activity-dependent modulation of synapse-regulating genes in astrocytes | https://www.ncbi.nlm.nih.gov/geo/query/acc.cgi?acc=GSE163775 | NCBI Gene Expression Omnibus, GSE163775 |

The following previously published datasets were used:

| Author(s) | Year | Dataset title | Dataset URL | Database and Identifier |
|---|---|---|---|---|
| Boisvert M, Allen N | 2017 | Astrocyte-enriched pan-brain aging transcriptome | https://www.ncbi.nlm.nih.gov/geo/query/acc.cgi?acc=GSE99791 | NCBI Gene Expression Omnibus, GSE99791 |

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
