## [Decision Letter]

**Acceptance summary:**

In this study the authors investigate the transcriptome and synaptogenic function of astrocytes in the developing visual cortex (VC), a widely used model for neural development. Using a combination of bulk RNAseq and detailed histology, the authors report the changing astrocyte transcriptome during VC development and show the expression of key synaptogenic genes are both timepoint and layer specific during development. This study provides an essential resource for understanding how astrocytes change and impact the development of VC circuits.The authors further demonstrate that neuronal cell-type and astrocyte interactions drive VC development and has implications for brain development. This study will be of broad interest to the fields of neuroscience and glial biology.

**Decision letter after peer review:**

Thank you for submitting your article "Activity-Dependent Modulation of Synapse-Regulating Genes in Astrocytes" for consideration by *eLife*. Your article has been reviewed by 3 peer reviewers, including Beth Stevens as the Reviewing Editor and Reviewer #1, and the evaluation has been overseen by Lu Chen as the Senior Editor.

Essential revisions:

This is an excellent paper that will provide a new framework for understanding how neurons shape developing astrocytes in the cortex. The experiments are well done and rigorous.

Overall, this is a very valuable study that needs some clarification and follow up that could be accomplished with a straightforward set of revisions without the necessity of new experiments. Please see specific reviewer comments, especially those highlighted below:

– Validation some of the top genes or layer specific genes shown in Figure 6 would strengthen the paper.

– The discussion could include a few sentences on the limitations of the study that could be addressed in future work (e.g., functional studies and validation with immunohistochemistry).

– Consider adding panels Figure 1SG and H to Figure 1 to round it out from the view of common and age specific GO terms and pathways? Also, in Figure 1F, how many genes were shared between developmental stages and how many were unique to each? Could this be shown near Figure 1F.

– For the scRNAseq analysis, it would be helpful for reader to display some of the astrocyte subclustering in a 2-dimensional fashion (eg. UMAP/tSNE) with the conditions and clusters labelled.

*Reviewer #1 (Recommendations for the authors):*

This paper is extremely impactful and rigorous. There are a few minor comments that would be worth fixing or clarifying:

1. Could the authors comment on the potential astrocyte coverage of Aldh1l1-Cre v Gfap-Cre? Why did the authors use the latter? Figure S1 does go into detail regarding specificity of this line, but I wonder how many of these stainings label astrocyte subtypes v pan astrocytes?

2. When describing the RNAseq results, could the authors avoid the term "most astrocyte specific" when describing eg. Lars2. Given the absence of other cell-types analyzed, maybe highly enriched would be more appropriate.

3. In the paper, the authors show beautiful GFP+smFISH images. However, the protocol section of paper does not appear to describe how they did this? In our experience RNAscope on fresh frozen tissue with the protease digestions described results in no endogenous GFP signal. Could the authors clarify this? If they developed a modified protocol it would be of immense use to the field.

4. For the scRNAseq analysis, it would be helpful for reader to display some of the astrocyte subclustering in a 2-dimensional fashion (eg. UMAP/tSNE) with the conditions and clusters labelled.

5. Finally give the extensive data in this paper and the lack of figure limits at *eLife*, could the authors create a summary schematic of their work and model? I think this would be very helpful for the reader.

*Reviewer #2 (Recommendations for the authors):*

This is an excellent paper that is appropriate for publication in *eLife*. Just a few minor points that need to be addressed:

1) In Figure 4 they manipulate neuronal activity and in parallel assess how the expression of synaptogenic genes expression. In this experiment, they should demonstrate that neuonral activity is actually altered. This can be done with field recordings with ephys or cFos staining of neurons. Also, have the authors considered a gain of neuronal activity experiment? The Vglut2-KO is a classic loss of neuronal activity paradigm, but it would be interesting to see what happened to these important genes with a gain of activity study---this could be achieved with a GqDREADD or ChR manipulation. I would predict that increased neuronal would have the opposite effect on these important genes.

2) They infer in figures 4/5 that astrocyte or neuronal activity regulates expression of GPC or Thbs family members, but there is no molecular or transcriptional mechanism pursued. This is totally fine, but I think a few sentences in the discussion that speculates about transcription factors regulate these genes and how their expression is linked to Ca^2+^ responses in astrocytes will help the reader.

3) The data in figure 6 is quite nice. However it seems like the authors should attempt to validate some top genes or layer specific genes. It would be great if they could unequivocally prove that some of these genes that came up in their profiling studies are in fact activity or layer specific.

*Reviewer #3 (Recommendations for the authors):*

Specific changes and clarifications:

1. Since the authors clearly showed distinct spatial transcriptional profiles of synapse-regulating genes in VC of VGlut 2 cKO and Ip3r2 cKO mice in Figure 4 and Figure 5 and investigated transcriptional changes in astrocytes of these mice by snRNA-seq in Figure 6 and Figure S9, it would be useful to pick two genes from Figure 6h (e.g. Sltm and Gm47283) that are clearly different from genotypes and validate them with FISH. If they have data along these lines they should add it. If they don't they could mention this as future hypothesis-driven work enabled by the current work.

2. In relation to Figure 6c, it would be useful to show gene expression of Gpc4, Gpc6, Chrdl1 in these cell clusters. This could be added to the sup info.

3. Figure 1, Figure S1: Do they have immunohistochemical analysis for cell-type specific expression of Rpl22-HA for P7 mice like they showed for P28? Or is the specificity based on RNA-seq from Figure S1D? Just a few clarification sentences needed.

4. Figure S1E: Rational for definition of astrocyte-specific genes (FPKM>100) is not clear compared with astrocyte-enriched genes. Please add a few clear sentences to the results about what is meant by astrocyte-specific versus astrocyte-enriched and how these two things were defined.

5. Figure S2A: Please clarify the cortical layer from which the image panels were gathered.

6. In Figure 4C, the vGlut1 expression in deep layers in VC looks lower in VGlut2 cKO mice, although the average data clearly show this is not the case. Could this be due to gray scale pixel values for the representative images? Please double check.

7. In Figure 4J and 4N, the colocalization of VGlut1/GluA or VGlut1/GluA2 is unclear. It is probably necessary to show zoomed-in images to make this cleaerer. This is also true for Figure 5I and 5M.

8. In Figure 4B, C what are the units of the x-axis? Is it VGlut2 signal intensity (a.u.) / um^2^? Something seems to be missing from the label. Also in Figure 4C.

9. On page 7 and in other places, when reporting data from FISH, best to show areas rounded up to 1 decimal place for areas rather than 2 decimal places? For small areas of a couple of um, it is unlikely that areas are accurate to two decimal places.

10. The discussion could include a paragraph discussing the limitations of the study that could be addressed in future work (e.g. functional studies and validation with immunohistochemistry).

11. One of the interesting findings from the work is that astrocytes and other cells from Ip3R2 ko mice display changes in synapse-regulating molecules as well as broad changes in gene expression (this latter point was also made recently in pubmed ID 33086039, and this could be cited but it is not critical to do so). This is important for the interpretation of studies where these mice have been used to either show or not show changes in mouse behavior from the angle of astrocyte calcium signals. The authors should add a couple of sentences to the discussion to indicate that behavioral changes may have been due to synaptogenic effects rather than astrocyte calcium signaling as such. This would be valuable to include as a discussion point.

12. I wonder if panels Figure 1SG and H could be added to Figure 1 to round it out from the view of common and age specific GO terms and pathways? Also in Figure 1F, how many genes were shared between developmental stages and how many were unique to each? Could this be shown near Figure 1F.

Overall, this is a very valuable study that needs a few more clarifications that could be accomplished with a straightforward set of revisions without the necessity of new experiments.

---

## [Author Response]

Essential revisions:This is an excellent paper that will provide a new framework for understanding how neurons shape developing astrocytes in the cortex. The experiments are well done and rigorous.Overall, this is a very valuable study that needs some clarification and follow up that could be accomplished with a straightforward set of revisions without the necessity of new experiments. Please see specific reviewer comments, especially those highlighted below:– Validation some of the top genes or layer specific genes shown in Figure 6 would strengthen the paper.

To validate this data, we took advantage of the publicly available in situ hybridization database of the developing mouse brain generated by the Allen Brain Institute (https://mouse.brain-map.org/). This dataset contains in situ hybridization images for a selection of genes important for brain development, including at P14, the time when our snRNAseq analysis was performed. Cross-referencing the putative layer-enriched genes against the developmental database identified several genes including Kcnd2, Id3 and Gfap, whose expression pattern in the VC matched their predicted layer localization in our dataset. These images are added to Figure 6—figure supplement 2B and text modified accordingly on lines 518-523.

– The discussion could include a few sentences on the limitations of the study that could be addressed in future work (e.g., functional studies and validation with immunohistochemistry).

We revised the Discussion section as suggested, and added future experiments to expand and support this study’s findings for each subsection, including:

Line 689: “Before performing functional studies based on these genes further characterization is required, for example cross-referencing these genes with our bulk RNA sequencing dataset to identify astrocyte-enriched genes, and performing immunohistochemistry to determine if protein is also heterogeneous.”

Line 702: “Functional studies are further needed to identify the precise neuronal activity patterns that govern astrocyte-neuron reciprocal communication.”

Line 739: “Future strategies including manipulation of neuronal and astrocyte function using opto- or chemogenetic approaches will further elucidate the role of astrocyte-neuron interaction in circuit development and maturation”.

Line 785: “Future studies employing functional approaches such as electrophysiology, optogenetic manipulations and behavior are needed to determine the precise nature of astrocyte plasticity and to further distinguish intrinsic and extrinsic influences on these cells giving further insight into their function in both health and disease.”

– Consider adding panels Figure 1SG and H to Figure 1 to round it out from the view of common and age specific GO terms and pathways? Also, in Figure 1F, how many genes were shared between developmental stages and how many were unique to each? Could this be shown near Figure 1F.

As suggested, we have added these panels to the main figure (Figure 1G, H), and created a new venn diagram showing the number of astrocyte enriched genes unique to each age, and common (Figure S1F new label: Figure 1—figure supplement 1F). In addition, we generated a heatmap showing the top 5 genes uniquely enriched in astrocytes at each age, which is presented in Figure 1F. The text has been modified accordingly (Lines 134-151).

– For the scRNAseq analysis, it would be helpful for reader to display some of the astrocyte subclustering in a 2-dimensional fashion (eg. UMAP/tSNE) with the conditions and clusters labelled.

Thank you for this suggestion. We have added UMAP plots showing the 4 astrocyte clusters for each genetic model to Figure 6C and Figure 6—figure supplement 2A.

Reviewer #1 (Recommendations for the authors):This paper is extremely impactful and rigorous. There are a few minor comments that would be worth fixing or clarifying:1. Could the authors comment on the potential astrocyte coverage of Aldh1l1-Cre v Gfap-Cre? Why did the authors use the latter? Figure S1 does go into detail regarding specificity of this line, but I wonder how many of these stainings label astrocyte subtypes v pan astrocytes?

We thank the reviewer for raising this important point. In our previous work, we characterized the Aldh1l1-cre line and found a robust neuronal recombination at P3 in the visual cortex (Farhy-Tselnicker et al. Neuron. FigS6^1^), making this line unsuitable for targeting astrocytes in this region during early postnatal development. In the same study, we compared two GFAP-cre lines, GFAP-cre 77.6 and GFAP-cre 73.12, and found that while both had high specificity for astrocytes at P3, they had differing efficiency, with the 77.6 line labeling far fewer astrocytes than the 73.12 line. This led us to choose the GFAP-cre 73.12 line for the Ribotag experiments in this study presented in Figure 1. As shown in Figure 1 —figure supplement 1C, greater than 95% of astrocytes in the visual cortex at P28 (marked by S100β) are also immuno-positive for HA-tagged ribosomes, demonstrating the majority of astrocytes are expressing the transgene showing high coverage. We have added a sentence highlighting this point into the Results section lines 109-113. In the case of the Aldh1l1-GFP line used for smFISH analysis of astrocyte genes in Figure 2, we performed immunostaining against a panel of astrocyte markers (Figure 2—figure supplement 1 S2A-C), finding a high overlap between GFP expressing cells and astrocyte markers. These results suggest high coverage of the astrocyte population in this mouse line, a point we have emphasized on lines 201-204.

2. When describing the RNAseq results, could the authors avoid the term "most astrocyte specific" when describing eg. Lars2. Given the absence of other cell-types analyzed, maybe highly enriched would be more appropriate.

Thank you for pointing out that this term could be misconstrued. We have updated the text accordingly (lines 155-158) and added the fold change level to emphasize the level of enrichment in astrocytes vs all cortical cells.

3. In the paper, the authors show beautiful GFP+smFISH images. However, the protocol section of paper does not appear to describe how they did this? In our experience RNAscope on fresh frozen tissue with the protease digestions described results in no endogenous GFP signal. Could the authors clarify this? If they developed a modified protocol it would be of immense use to the field.

We thank the reviewer for this comment, and apologize for the lack of clarity in the methods section. The smFISH was performed on 4% PFA fixed tissue that was processed using cryo-sectioning to generate brain sections. We then followed the ACDbio protocol for fixed-frozen tissue with some modifications. We found this method to be very robust for young mice (up to P28) used in this study. For older ages an antigen retrieval step is required, which quenches the endogenous GFP signal. We have revised and expanded this part of the methods section under “single-molecule fluorescent in situ hybridization (smFISH) (line 1132) to further explain the modifications used.

4. For the scRNAseq analysis, it would be helpful for reader to display some of the astrocyte subclustering in a 2-dimensional fashion (eg. UMAP/tSNE) with the conditions and clusters labelled.

Thank you for this suggestion. We have added UMAP plots showing the 4 astrocyte clusters for each genetic model to Figure 6C and Figure 6—figure supplement 2A.

5. Finally give the extensive data in this paper and the lack of figure limits at eLife, could the authors create a summary schematic of their work and model? I think this would be very helpful for the reader.

We thank the reviewer for this excellent suggestion. We created an additional figure (Figure 7) containing the schematic of the work.

Reviewer #2 (Recommendations for the authors):This is an excellent paper that is appropriate for publication in eLife. Just a few minor points that need to be addressed:1) In Figure 4 they manipulate neuronal activity and in parallel assess how the expression of synaptogenic genes expression. In this experiment, they should demonstrate that neuonral activity is actually altered. This can be done with field recordings with ephys or cFos staining of neurons. Also, have the authors considered a gain of neuronal activity experiment? The Vglut2-KO is a classic loss of neuronal activity paradigm, but it would be interesting to see what happened to these important genes with a gain of activity study---this could be achieved with a GqDREADD or ChR manipulation. I would predict that increased neuronal would have the opposite effect on these important genes.

We thank the reviewer for these insightful comments. Diminishing neuronal activity via VGlut2 knockout is an established paradigm that has been validated in other studies, as cited in the manuscript (lines 329-331), prompting us to use this same approach for our study. However, we agree that determining the specific change in neuronal activity pattern in the visual cortex in the absence of thalamic glutamate release, and how this regulates astrocyte gene expression, is an important next step. We have added a sentence to the Discussion explaining this point, lines 702-703.

Concerning gain of neuronal activity, we completely agree with the reviewer’s prediction. This is further supported by our experiments using cultured neurons and astrocytes, which show a decrease in Gpc4 protein secretion from astrocytes co-cultured with neurons compared to cultured alone (Figure 4—figure supplement 1A). To investigate if this is due to neurotransmitters acting on astrocytes, and thus linked to neuronal activity, we performed an additional set of experiments where we treated cultured astrocytes with neurotransmitters to mimic increased neuronal activity (new Figure 4—figure supplement 1B). These data show that Gpc4 secretion from astrocytes is also decreased in the presence of the neuro-transmitters glutamate, adenosine or ATP (Figure 4—figure supplement 1B; line 315). This suggests, that in vivo, the decrease observed in Gpc4 expression at P14 stems from alterations in the neuronal activity pattern between P7 and P14. Future work employing neuronal and astrocyte activation in vivo using the methods suggested by the reviewer is needed to further expand on these findings. We have added sentences to the Discussion about this point, lines 739, 785.

2) They infer in figures 4/5 that astrocyte or neuronal activity regulates expression of GPC or Thbs family members, but there is no molecular or transcriptional mechanism pursued. This is totally fine, but I think a few sentences in the discussion that speculates about transcription factors regulate these genes and how their expression is linked to Ca^2+^ responses in astrocytes will help the reader.

We thank the reviewer for this important comment. We revised the discussion to include these points (lines 757-760).

3) The data in figure 6 is quite nice. However it seems like the authors should attempt to validate some top genes or layer specific genes. It would be great if they could unequivocally prove that some of these genes that came up in their profiling studies are in fact activity or layer specific.

To validate this data, we took advantage of the publicly available in situ hybridization database of the developing mouse brain generated by the Allen Brain Institute (https://mouse.brain-map.org/). This dataset contains in situ hybridization images for a selection of genes important for brain development, including at P14, the time when our snRNAseq analysis was performed. Cross-referencing the putative layer-enriched genes against the developmental database identified several genes including Kcnd2, Id3 and Gfap, whose expression pattern in the VC matched their predicted layer localization in our dataset. These images are added to Figure 6—figure supplement 2B and text modified accordingly on lines 518-523.

Reviewer #3 (Recommendations for the authors):Specific changes and clarifications1. Since the authors clearly showed distinct spatial transcriptional profiles of synapse-regulating genes in VC of VGlut 2 cKO and Ip3r2 cKO mice in Figure 4 and Figure 5 and investigated transcriptional changes in astrocytes of these mice by snRNA-seq in Figure 6 and Figure S9, it would be useful to pick two genes from Figure 6h (e.g. Sltm and Gm47283) that are clearly different from genotypes and validate them with FISH. If they have data along these lines they should add it. If they don't they could mention this as future hypothesis-driven work enabled by the current work.

We thank the reviewer for this important comment. We agree that upon validation, these data may form the basis of important functional studies around the roles of activity-regulated genes in astrocytes. We have updated the discussion to emphasize these points and to discuss potential future directions (line 689; line 774).

2. In relation to Figure 6c, it would be useful to show gene expression of Gpc4, Gpc6, Chrdl1 in these cell clusters. This could be added to the sup info.

These data are shown as dot plots in Figure 6—figure supplement 2I. Unfortunately, the sensitivity of the snRNAseq assay resulted in low detection of Gpc4 and Chrdl1 genes. Nevertheless, we plotted total CPM values of these genes as a heatmap, and found that their expression levels in each cluster match up to our in situ hybridization data at P14, namely, Gpc4 expression is lowest in the upper layer cluster, Chrdl1 expression is highest in the upper layer cluster, and Gpc6 expression is similar across all clusters. While the lack of statistical significance precludes making strong conclusions based on these data, the trend and similarity to the smFISH results are reassuring. The heatmap is shown in Figure 6—figure supplement 2J, and text edited accordingly (lines 551-557).

3. Figure 1, Figure S1: Do they have immunohistochemical analysis for cell-type specific expression of Rpl22-HA for P7 mice like they showed for P28? Or is the specificity based on RNA-seq from Figure S1D? Just a few clarification sentences needed.

The specificity at P7 is based on the RNAseq data from Figure S1D (moved to Figure 1—figure supplement 1A in the revised version of the manuscript). While we did obtain IHC example images showing HA tag expression in astrocytes for P7 (as shown in Figure 1B), quantification was only performed at P28. To clarify this, we edited the text accordingly (lines 107-113), and edited the figure to show the RNA-seq data first, and IHC data second.

4. Figure S1E: Rational for definition of astrocyte-specific genes (FPKM>100) is not clear compared with astrocyte-enriched genes. Please add a few clear sentences to the results about what is meant by astrocyte-specific versus astrocyte-enriched and how these two things were defined.

We have edited the text to clarify this point and removed the wording “astrocyte specific” as suggested by Reviewer 1 (lines 155-158). In this dataset genes with an FPKM>100 were ranked by their enrichment score (FPKM in astrocyte/all cells i.e. input). These criteria were chosen to emphasize genes that are robustly expressed by and enriched in astrocytes.

5. Figure S2A: Please clarify the cortical layer from which the image panels were gathered.

The images were taken from mid cortical layers L2-4. We added this information to the panel header in Figure 1—figure supplement 1D.

6. In Figure 4C, the vGlut1 expression in deep layers in VC looks lower in VGlut2 cKO mice, although the average data clearly show this is not the case. Could this be due to gray scale pixel values for the representative images? Please double check.

We thank the reviewer for pointing this out. Upon inspection of the representative images we found that indeed, the gray scale values were not the same in the two images. We corrected this issue, as can be seen in the revised image in Figure 4C.

7. In Figure 4J and 4N, the colocalization of VGlut1/GluA or VGlut1/GluA2 is unclear. It is probably necessary to show zoomed-in images to make this cleaerer. This is also true for Figure 5I and 5M.

We have added a new panel showing zoomed in colocalization images of synaptic markers in Figures 4, 5, Figure 4—figure supplement 2 and Figure 5—figure supplement 1. For images in Figure 4—figure supplement 2 which show triple colocalization, we changed the pseudo-color of Bassoon to green to further enhance visibility of colocalization.

8. In Figure 4B,C what are the units of the x-axis? Is it VGlut2 signal intensity (a.u.) / um^2^? Something seems to be missing from the label. Also in Figure 4C.

We apologize for this oversight, and have corrected the X axis label in both Figure 4B and C to the correct units: thresholded area per um^2^.

9. On page 7 and in other places, when reporting data from FISH, best to show areas rounded up to 1 decimal place for areas rather than 2 decimal places? For small areas of a couple of um, it is unlikely that areas are accurate to two decimal places.

We have rounded all the smFISH data presented in the manuscript to 1 decimal place.

10. The discussion could include a paragraph discussing the limitations of the study that could be addressed in future work (e.g. functional studies and validation with immunohistochemistry).

We revised the Discussion section as suggested, and added future experiments to expand and support this study’s findings for each subsection, including:

Line 689: “Before performing functional studies based on these genes further characterization is required, for example cross-referencing these genes with our bulk RNA sequencing dataset to identify astrocyte-enriched genes, and performing immunohistochemistry to determine if protein is also heterogeneous.”

Line 702: “Functional studies are further needed to identify the precise neuronal activity patterns that govern astrocyte-neuron reciprocal communication.”

Line 739: “Future strategies including manipulation of neuronal and astrocyte function using opto- or chemogenetic approaches will further elucidate the role of astrocyte-neuron interaction in circuit development and maturation”.

Line 785: “Future studies employing functional approaches such as electrophysiology, optogenetic manipulations and behavior are needed to determine the precise nature of astrocyte plasticity and to further distinguish intrinsic and extrinsic influences on these cells giving further insight into their function in both health and disease.”

11. One of the interesting findings from the work is that astrocytes and other cells from Ip3R2 ko mice display changes in synapse-regulating molecules as well as broad changes in gene expression (this latter point was also made recently in pubmed ID 33086039, and this could be cited but it is not critical to do so). This is important for the interpretation of studies where these mice have been used to either show or not show changes in mouse behavior from the angle of astrocyte calcium signals. The authors should add a couple of sentences to the discussion to indicate that behavioral changes may have been due to synaptogenic effects rather than astrocyte calcium signaling as such. This would be valuable to include as a discussion point.

We thank the reviewer for raising this important point. It is clear that Ip3r2 is crucial for mediating multiple types of astrocyte function, the breadth of which are just now beginning to be elucidated. We have added this point to the Discussion section on lines 764-769 of the manuscript. We also added the suggested citation.

12. I wonder if panels Figure 1SG and H could be added to Figure 1 to round it out from the view of common and age specific GO terms and pathways? Also in Figure 1F, how many genes were shared between developmental stages and how many were unique to each? Could this be shown near Figure 1F.

As suggested, we have added these panels to the main figure (Figure 1G, H), and created a new venn diagram showing the number of astrocyte enriched genes unique to each age, and common (Figure S1F new label: Figure 1—figure supplement 1F). In addition, we generated a heatmap showing the top 5 genes uniquely enriched in astrocytes at each age, which is presented in Figure 1F. The text has been modified accordingly (Lines 134-151). References

1 Farhy-Tselnicker, I. *et al.,* Astrocyte-Secreted Glypican 4 Regulates Release of Neuronal Pentraxin 1 from Axons to Induce Functional Synapse Formation. *Neuron* 96, 428-445.e413, doi:10.1016/j.neuron.2017.09.053 (2017).